# Interference Among First-Price Pacing Equilibria: A Bias and Variance Analysis

**Luofeng Liao**   **Christian Kroer**
Columbia University
{ll3530, ck2945}@columbia.edu

**Sergei Leonenkov**
Ads Online Experimentation, Meta
leonenkov@meta.com

**Okke Schrijvers**   **Liang Shi**   **Nicolas Stier-Moses**   **Congshan Zhang**
Central Applied Science, Meta
{okke, liangshi, nstier, cszhang}@meta.com

## Abstract

A/B testing is widely used in the internet industry. For online marketplaces (such as advertising markets), standard approaches to A/B testing may lead to biased results when buyers have budget constraints, as budget consumption in one arm of the experiment impacts performance of the other arm. This is often addressed using a budget-split design. Yet such splitting may degrade statistical performance as budgets become too small in each arm. We propose a *parallel budget-controlled A/B testing* design where we use market segmentation to identify submarkets in the larger market, and we run parallel budget-split experiments in each submarket. We demonstrate the effectiveness of this approach on real experiments on advertising markets at Meta. Then, we formally study interference that derives from such experimental designs, using the first-price pacing equilibrium framework as our model of market equilibration. We propose a debiased surrogate that eliminates the first-order bias of FPPE, and derive a plug-in estimator for the surrogate and establish its asymptotic normality. We then provide an estimation procedure for submarket parallel budget-controlled A/B tests. Finally, we present numerical examples on semi-synthetic data, confirming that the debiasing technique achieves the desired coverage properties.

## 1 Introduction

Online A/B testing is widely used in the internet industry to inform decisions on new feature roll-outs. For online marketplaces (such as advertising markets), standard approaches to A/B testing may lead to biased results when buyers operate under a budget constraint, as budget consumption in one arm of the experiment impacts performance of the other arm. To counteract this interference, one can use a budget-split design where the budget constraint operates on a per-arm basis and each arm receives an equal fraction of the budget, leading to "budget-controlled A/B testing," see e.g. Basse et al. (2016); Liu et al. (2021).

Despite clear advantages of budget-controlled A/B testing, companies are extremely constrained by the number of such experiments they can run. While it's possible to create more budget splits, this will lower the budget per group substantially, which could lead to different equilibrium outcomes and may disproportionately affect smaller buyers. Additionally, a common approach to increase experimentation throughput is to run orthogonal experiments (with their own orthogonal randomization), but this would either suffer from the same interference as the vanilla A/B test setup, or also require further budget splits.

In this paper, we propose a *parallel budget-controlled A/B test* design where we use market segmentation to identify submarkets in the larger market, and we run parallel experiments on each submarket. When the overall market can be divided into several relatively isolated submarkets, budget-controlled A/B tests can be conducted in parallel within these submarkets. However, this method also presents some challenges. First, submarkets are rarely completely isolated; certain items may

attract buyers from multiple submarkets, resulting in interference across submarkets when conducting tests in parallel. Second, submarkets differ in terms of buyer (and user) composition, which might cause the local treatment effect estimates to not be representative of the global treatment effect where all buyers are included in the market. The second challenge is relatively easy to address in practice by imposing balancing constraints in the clustering algorithm used to define submarkets, while the first challenge is more fundamentally important and requires deeper understanding.

Before the theoretical exposition, we consider a comparison of results for paired experiments between a parallel budget-controlled A/B test setup, and that of a traditional budget-split design; where the latter is considered the gold standard. Sec. 1 on the left shows comparisons of 99 experiments where the point estimate and CIs are plotted on the vertical axis for the parallel design, and on the horizontal axis for the budget-split design. The most important feature is whether the two experiments agree between (negative, neutral, positive), as a change would result in a launch reversal. The two experiment designs agree in 75% of cases (at 90% confidence level, hence the optimal agreement is 81.5%), which increases to 79% after the introducing a guardrail metric, see Sec. 1 on the right. These results are quite satisfactory, but do point at the existence of remaining interference bias. In the remainder of this paper, correcting the interference bias is the main objective.

**Contributions:**

1. We formally define market interference in first-price auction markets using the first price pacing equilibrium (FPPE) framework (Conitzer et al., 2022a) (Sec. 3)

2. We propose a debiased surrogate that eliminates the first-order bias of FPPE, and derive a plug-in estimator for the surrogate and establish its asymptotic normality. (Sec. 4)

3. We run semi-synthetic experiments, confirming that the debiasing technique achieves the desired coverage properties. (Sec 5).

## 1.1 PARALLEL A/B TESTING IN PRACTICE

In this section we describe the real-world problem of A/B testing with congestion that we wish to model, and our proposed solution of parallel A/B tests in carefully balanced submarkets. We start by describing the market environment. There is a set of $n$ advertisers, and each advertiser $i$ has a budget $b_i$. Whenever a user shows up on the platform an *impression opportunity* occurs, and an auction is conducted in order to determine which ad will be shown to the user. Each advertiser $i$ has some stated value $v_i(\theta)$ of being shown to a particular impression opportunity $\theta$. The advertiser submits a bid which is determined based on $v_i(\theta)$, as well as the expenditure of the advertiser so far. For example, in *multiplicative pacing* Balseiro et al. (2017); Conitzer et al. (2022b), the platform adaptively learns a *pacing multiplier* $\beta_i \in [0, 1]$ such that the bid is formed as $\beta_i v_i(\theta)$. The budget-management system then adaptively controls $\beta_i$ over time, in order to ensure the correct rate of budget expenditure on behalf of the advertiser. We consider first-price auctions, which is the predominant way display advertising is sold online.

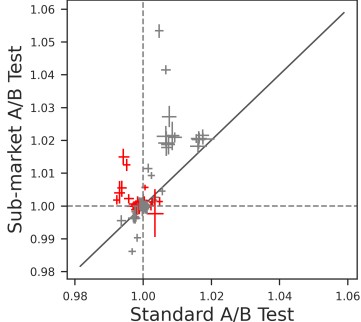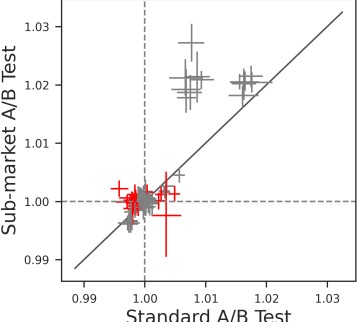

**Figure 1:** Parallel vs. standard budget-controlled A/B test, daily treatment effect. We denote neutral treatment effects with a value of 1.0. Red crosses indicate instances of sign inconsistencies. Left are all datapoints, on the right, datapoints that fail a guardrail metric are removed.

Now that we have discussed budget management, we describe the A/B testing problem. Suppose that a platform wants to run $K$ A/B tests, which may affect, e.g., the valuations that advertisers have for impression slots, revenue, etc. We construct the market-segmented experimental setup as follows: We first define a bipartite graph between advertisers and users based on targeting criteria. Subsequently, we cluster the advertisers into $K$ clusters, where the objective is to minimize the sum of weighted edges between clusters subject to traffic balancing constraints to make the resulting clusters as similar to the whole market as possible. The edge weight between a pair of advertisers is the number of impressions (or users) where they are both within the top-$k$ bids. The choice of $k$ is a parameter that must be chosen based on experience with the specific application setting. If the clustering achieves a small objective function value, then each cluster is a mostly isolated submarket, in the sense that each user will mostly receive bids from advertisers in a single cluster. Then, we run an A/B test within each of the $K$ submarkets. Every user is randomly assigned to either "A" or "B" in each submarket. The main challenge is that while submarkets are relatively isolated, there is remaining interference from users who are targeted by advertisers from different submarkets, leading to a slightly different equilibrium. Our main contribution is to define a framework for analyzing such interference, and giving an estimator that removes the bias from these users. We survey related works in App. D.

## 2 REVIEW OF FPPE THEORY

**Notation**. For a measurable space $(\Theta, \mathrm{d}\theta)$, we let $L^p$ (and $L^p_+$, resp.) denote the set of (nonnegative, resp.) $L^p$ functions on $\Theta$ w.r.t the base measure $\mathrm{d}\theta$ for any $p \in [1, \infty]$ (including $p = \infty$). Given $x \in L^\infty$ and $v \in L^1$, we let $\langle v, x \rangle = \int_\Theta v(\theta)x(\theta)\,\mathrm{d}\theta$. We treat all functions that agree on all but a measure-zero set as the same. For a sequence of random variables $\{X_n\}$, we say $X_n = O_p(1)$ if for any $\epsilon > 0$ there exists a finite $M_\epsilon$ and a finite $N_\epsilon$ such that $\mathbb{P}(|X_n| > M_\epsilon) < \epsilon$ for all $n \geq N_\epsilon$. We say $X_n = o_p(1)$ if $X_n$ converges to zero in probability. For a subset $\Theta' \subset \Theta$, let $1_{\Theta'}(\cdot) : \Theta \to \{0, 1\}$ be the indicator function of $\Theta'$. Convergence in distribution and probability is denoted by $\xrightarrow{d}$ and $\xrightarrow{p}$. Given a vector $a = [a_1, \ldots, a_n]^\top$, let $\mathrm{Diag}(a)$ denote the diagonal matrix with $(i, i)$-th entry being $a_i$; sometimes we write $\mathrm{Diag}(a_i)$ when it is convenient to define each $a_i$ inline. Let $\boldsymbol{A}^\dagger$ denote the Moore–Penrose inverse of the matrix $\boldsymbol{A}$, $\boldsymbol{e}_j$ the $j$-th unit vector, and $[n] = \{1, \ldots, n\}$.

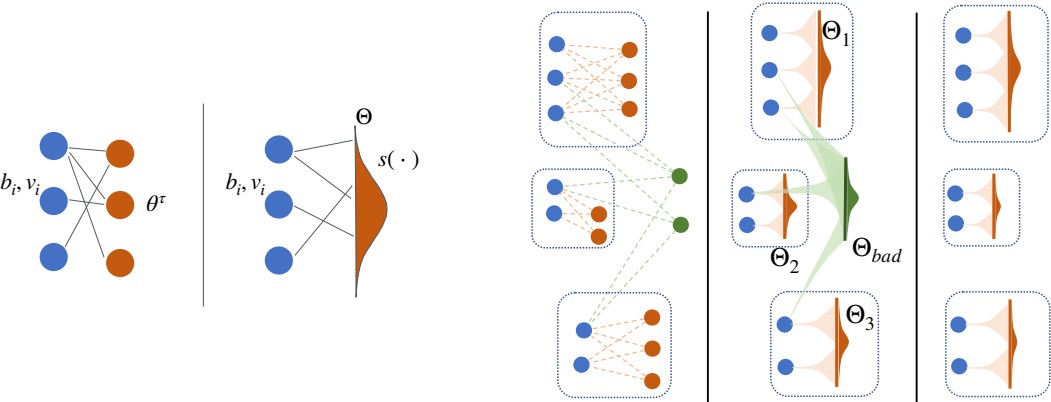

**Figure 2: Left:** Finite FPPE (left) and limit FPPE (right). In a finite FPPE, there are a finite number of items; in a limit FPPE, the item set is a continuum. **Right:** The interference model — Left ($\widehat{\mathscr{M}_\alpha}$): the observed market where interference is present among submarkets. Middle ($\mathscr{M}_\alpha$): the limit market with interference from bad item set. Right ($\mathscr{M}_0$): the limit market with perfectly separated submarkets. We use data from the left panel to make inferences about the market in the right panel.

**Limit FPPE**. We first introduce our notion of a limit market and two regularity conditions on the market, which models the underlying market structure that we sample from. We have $n$ buyers and a possibly continuous set of items $\Theta$ with an integrating measure $\mathrm{d}\theta$. For example, one could take $\Theta =$

$[0, 1]$ and $\mathrm{d}\theta$ = the Lebesgue measure on $[0, 1]$. Each buyer has a *budget* $b_i$; let $\boldsymbol{b} = (b_1, \ldots, b_n)$. The *valuation* for buyer $i$ is a function $v_i \in L^1_+$ such that buyer $i$ has valuation $v_i(\theta)$ for one unit of item $\theta \in \Theta$; let $\boldsymbol{v} : \Theta \to \mathbb{R}^n$, $\boldsymbol{v}(\theta) = [v_1(\theta), \ldots, v_n(\theta)]$. We assume $\bar{v} = \max_i \sup_\theta v_i(\theta) < \infty$. The *supplies* of items are given by a function $s \in L^\infty_+$, i.e., item $\theta \in \Theta$ has $s(\theta)$ units of supply. Without loss of generality, we assume a unit total supply $\int_\Theta s \, \mathrm{d}\theta = 1$. Given $g : \Theta \to \mathbb{R}$, we let $\mathbb{E}[g] = \int g(\theta)s(\theta) \, \mathrm{d}\theta$ and $Var[g] = \mathbb{E}[g^2] - (\mathbb{E}[g])^2$. Given $t$ i.i.d. draws $\{\theta^1, \ldots, \theta^t\}$ from $s$, let $P_t g(\cdot) = \frac{1}{t}\sum_{\tau=1}^t g(\theta^\tau)$.

Next we introduce the market equilibrium concept that is the foundation of our study. For that we leverage the *first-price pacing equilibrium* (FPPE) (Conitzer et al., 2022a). FPPE models equilibrium outcomes under budget-management systems employed in several practical settings. Each buyer is assigned a *pacing multiplier* $\beta_i$, which is used to control their budget expenditure. For each individual auction $\theta$, the buyer bids $\beta_i v_i(\theta)$, which can be seen as their adjusted valuation after factoring in their budget constraint (in practice, the valuation $v_i(\theta)$ may not be the buyer's true valuation, but instead their unscaled bid reported to the platform). The goal of the budget management system is to achieve an equilibrium, meaning that it must ensure that buyers spend their budget exactly by appropriately choosing $\beta_i$. If the budget cannot be fully spent, then no pacing must occur (i.e., $\beta_i = 1$). Below we formally define the pacing equilibrium concept in the continuous setting (see Fig 2, right), and give the finite version in the next section (Fig 2, left).

**Definition 1** (Limit FPPE, Gao & Kroer (2022); Conitzer et al. (2022a))**.** *A limit FPPE, denoted as* FPPE($\boldsymbol{b}, \boldsymbol{v}, s, \Theta$)*, is the unique tuple* $(\boldsymbol{\beta}, p(\cdot)) \in [0, 1]^n \times L^1_+(\Theta)$ *such that there exist an allocation* $x_i : \Theta \to [0, 1]$ *for all* $i \in [n]$ *such that,*

1. *(First-price) Prices and allocations are determined by first price auctions: for all items* $\theta \in \Theta$, $p(\theta) = \max_i \beta_i v_i(\theta)$, *and only the highest bidders obtain items: for all* $i$ *and* $\theta$, $x_i(\theta) > 0$ *implies* $\beta_i v_i(\theta) = \max_k \beta_k v_k(\theta)$

2. *(Feasiblity, market clearing) Budget are respected: for all* $i$, $\int x_i(\theta)p(\theta)s(\theta) \, \mathrm{d}\theta \le b_i$. *There is no overselling: for all* $\theta$, $\sum_{i=1}^n x_i(\theta) \le 1$. *Items with nonzero price are fully allocated: for all* $\theta$, $p(\theta) > 0$ *implies* $\sum_{i=1}^n x_i(\theta) = 1$.

3. *(No unnecessary pacing) For all* $i$, $\int x_i(\theta)p(\theta)s(\theta) \, \mathrm{d}\theta < b_i$ *implies* $\beta_i = 1$.

Let $\boldsymbol{\beta}^*$ and $p^*$ be the equilibrium pacing multipliers and prices. Revenue in the limit FPPE is $REV^* = \int p^*(\theta)s(\theta) \, \mathrm{d}\theta$. It measures the profitability of the auction platform. The leftover budgets for buyers are denoted by $\delta_i^* = b_i - \int p^*(\theta)s(\theta)x_i^*(\theta) \, \mathrm{d}\theta$.

The first two conditions simply describe the possible outcomes of a first-price auction system that uses pacing as the budget management strategy. The last condition, no unnecessary pacing, ensures that we only scale down a buyer's bids in case their budget constraint is binding. FPPE has many nice properties, including that they are competitive equilibria and that they are revenue-maximizing among budget-feasible pacing multipliers (Conitzer et al., 2022a).

In a pacing auction market $\mathscr{M} = $ FPPE($\boldsymbol{b}, \boldsymbol{v}, s, \Theta$) the following two regularity conditions are important for the study of its statistical properties.

**Definition 2** (SMO)**.** *We say the smoothness condition holds if the map* $\boldsymbol{\beta} \mapsto \mathbb{E}_s[\max_i \beta_i v_i(\theta)]$ *is twice continuously differentiable in a neighborhood of* $\boldsymbol{\beta}^*$.

**Definition 3** (SCS)**.** *We say strict complementary slackness holds if, whenever a buyer is unpaced* ($\beta_i^* = 1$)*, then her leftover budget is strictly positive* ($\delta_i^* > 0$)*.*

The condition SMO ensures that in the limit market, items that incur a tie are measure zero. The condition SCS rules out degenerate buyers that spend their budget exactly at $\beta_i^*$. See Liao & Kroer (2023) and Liao et al. (2023) for an extensive discussion about these conditions.

**Finite FPPE.** Next we introduce the finite FPPE, which models the auction data we observe in practice. Let $\gamma = (\theta^1, \ldots, \theta^t)$ be a sequence of items. Assume each item has the same supply of $\sigma \in \mathbb{R}_+$ units. A finite FPPE is a limit FPPE where the supply is a discrete measure supported on the observed items $\gamma$. Let $v_i^\tau = v_i(\theta^\tau)$.

**Definition 4** (Finite FPPE, informal)**.** *A finite FPPE,* $\widehat{FPPE}(\boldsymbol{b}, \boldsymbol{v}, \sigma, \gamma)$*, is a limit FPPE where the item set is the finite set of observed items* $\gamma$*. See App. C for the full definition.*

In Liao & Kroer (2023), it is shown that if $\gamma$ consists of $t$ i.i.d. draws from distribution $s$, and one takes $\sigma = 1/t$, then the pacing multiplier in $\widehat{\mathsf{FPPE}}(\boldsymbol{b}, \boldsymbol{v}, 1/t, \gamma)$ converge to the pacing multiplier in $\mathsf{FPPE}(\boldsymbol{b}, \boldsymbol{v}, s, \Theta)$ in probability. Also, note that the FPPE $\widehat{\mathsf{FPPE}}(t\boldsymbol{b}, \boldsymbol{v}, 1, \gamma)$ converges to the same limit FPPE $\mathsf{FPPE}(\boldsymbol{b}, \boldsymbol{v}, s, \Theta)$ because the pacing multipliers of a finite FPPE do not change when budgets and supplies are multiplied by the same scalar.

**The Eisenberg-Gale Program**. Both the limit FPPE and the finite FPPE have convex program characterizations (Chen et al., 2007; Conitzer et al., 2022a; Gao & Kroer, 2022). We define the dual Eisenberg-Gale (EG) objective for a single item $\theta$ as

$$F(\theta, \boldsymbol{\beta}) = f(\theta, \boldsymbol{\beta}) - \sum_{i=1}^{n} b_i \log \beta_i , \quad f(\theta, \boldsymbol{\beta}) = \max_{i \in [n]} \beta_i v_i(\theta) . \tag{1}$$

The population and sample (dual) EG objectives are then defined as

$$H(\boldsymbol{\beta}) = \mathbb{E}[F(\theta, \boldsymbol{\beta})] , \quad H_t(\boldsymbol{\beta}) = \frac{1}{t} \sum_{\tau=1}^{t} F(\theta^\tau, \boldsymbol{\beta}) . \tag{2}$$

We say $\boldsymbol{H}$ is the Hessian of market $\mathscr{M}$ when $\boldsymbol{H} = \nabla^2_{\boldsymbol{\beta\beta}} \int F(\theta, \boldsymbol{\beta}) s \, \mathrm{d}\theta |_{\boldsymbol{\beta}=\boldsymbol{\beta}^*}$.

The equilibrium pacing multipliers $\boldsymbol{\beta}^*$ in $\mathsf{FPPE}(\boldsymbol{b}, \boldsymbol{v}, s, \Theta)$ can be recovered through the population dual EG program

$$\boldsymbol{\beta}^* = \arg\min_{\boldsymbol{\beta} \in (0,1]^n} H(\boldsymbol{\beta}) . \tag{3}$$

The pacing multiplier vector $\boldsymbol{\beta}^*$ is the unique solution to Eq. (3). Let $\boldsymbol{\beta}^\gamma$ be the equilibrium pacing multiplier in $\widehat{\mathsf{FPPE}}(\boldsymbol{b}, \boldsymbol{v}, 1/t, \gamma)$. Then $\boldsymbol{\beta}^\gamma$ solves the sample analogue of Eq. (3):

$$\boldsymbol{\beta}^\gamma = \arg\min_{\boldsymbol{\beta} \in (0,1]^n} H_t(\boldsymbol{\beta}) . \tag{4}$$

Let us briefly comment on the differential structure of $f$, since it plays a role in later sections. The function $f(\boldsymbol{\beta}, \theta)$ is a convex function of $\boldsymbol{\beta}$ and its subdifferential $\partial_{\boldsymbol{\beta}} f(\boldsymbol{\beta}, \theta)$ is the convex hull of $\{v_i \boldsymbol{e}_i : \text{index } i \text{ such that } \beta_i v_i(\theta) = \max_k \beta_k v_k(\theta)\}$, with $\boldsymbol{e}_i$ being the base vector in $\mathbb{R}^n$. When $\max_i \beta_i v_i(\theta)$ is attained by a unique $i^*$, the function $f(\cdot, \theta)$ is differentiable. In that case, all entries of $\nabla_{\boldsymbol{\beta}} f(\boldsymbol{\beta}, \theta)$ are zero expect that the $i^*$-th entry is filled with the value $v_{i^*}(\theta)$.

## 3 INTERFERENCE AS CONTAMINATION

In this section, we discuss how to estimate market equilibria when there is *contamination* in the supply, meaning that items are generated from a mixture of two distributions, when in reality we wish to estimate equilibrium quantities from one of the two distributions. Then, we show that interference from other markets can be viewed as a form of contamination, and so the problem of removing interference bias can be analyzed via our contamination framework.

### 3.1 FPPE WITH CONTAMINATED SUPPLY

We assume that we are in the same FPPE setting as before: there are $n$ buyers, each with budget $b_i$, and an item set $\Theta$ which is now partitioned in to $\Theta_{\mathsf{bad}}$ and $\Theta_{\mathsf{good}}$. However, now we assume that the supply $s$ is contaminated by $s'$, another supply distribution. We define the $\alpha$-contaminated market as $\mathscr{M}_\alpha = \mathsf{FPPE}(\boldsymbol{b}, \boldsymbol{v}, s_\alpha, \Theta)$ and the uncontaminated market as $\mathscr{M}_0 = \mathsf{FPPE}(\boldsymbol{b}, \boldsymbol{v}, s, \Theta)$, where $s_\alpha = \alpha s' + (1 - \alpha)s$, distribution $s'$ is supported on $\Theta_{\mathsf{bad}}$, and $s$ on $\Theta_{\mathsf{good}}$. Our goal is to perform inference about FPPE properties in the limit FPPE with the supply $s$. However, we are given access to finite FPPEs sampled from $s_\alpha$ instead. In particular, let $\gamma$ be $t$ i.i.d. draws from $s_\alpha$ and let $\widehat{\mathscr{M}_\alpha} = \widehat{\mathsf{FPPE}}(\boldsymbol{b}, \boldsymbol{v}, 1/t, \gamma)$. We assume $\alpha$ is known throughout the paper. In practice, this can often be estimated from historical data; in the parallel A/B test setting, this can be estimated directly from the sampled set of items, since we know whether an item is drawn from $s$ or $s'$. Let $\boldsymbol{\beta}^*$ and $\boldsymbol{\beta}^*_\alpha$ be the limit pacing multipliers in $\mathscr{M}_0$ and $\mathscr{M}_\alpha$, respectively. Let $\boldsymbol{\beta}^\gamma_\alpha$ be the pacing multipliers in the sampled market $\widehat{\mathscr{M}_\alpha}$ and let $H_{\alpha,t}(\boldsymbol{\beta}) = \frac{1}{t} \sum_{\tau=1}^{t} F(\theta^\tau, \boldsymbol{\beta})$ be the sample EG objective.

If we wanted to make inferences about $\mathcal{M}_\alpha$ then we could use existing statistical inference theory on how to use data in a finite FPPE $\widehat{\mathcal{M}_\alpha}$ to make inferences about the limit FPPE $\mathcal{M}_\alpha$ (Liao & Kroer, 2023). However, the supply contamination prevents the application of these tools to our problem.

Our central research question is then on how to use data in the finite contaminated market $\widehat{\mathcal{M}_\alpha}$ to make inferences about the uncontaminated limit market $\mathcal{M}_0$.

In Sec. 4 we propose an estimator for this problem and derive its properties. The results there apply to general item space $\Theta$ and supplies $s$ and $s'$. By imposing structure on $\Theta$, $s$ and $s'$, we show that the contamination model captures the interference among FPPEs.

## 3.2 Application: Modeling Interference among FPPEs

Now we show how the contamination model from the previous section can be used to model interference. Consider $K$ separate auction markets, which together form a global market. In the global market there are $n$ buyers, each with budget $b_i$, and an item set $\Theta$, partitioned into $\Theta_{\text{good}}$ and $\Theta_{\text{bad}}$. Let $C_1, \ldots, C_K$ be a partition of the buyers, $\Theta_1, \ldots, \Theta_K$ be a partition of the good item set $\Theta_{\text{good}}$, and $s_1, \ldots s_K$ be a set of supply functions, supported on $\Theta_1, \ldots, \Theta_K$ respectively. The $k$-th submarket consists of buyers in $C_k$, the item set $\Theta_k$ and supply $s_k$. Let $s = \frac{1}{K} \sum_k s_k$ be the average mixture and $s'$ be a supply supported on $\Theta_{\text{bad}}$. Let the contaminated supply be $s_\alpha = \alpha s' + (1-\alpha)s$.

By imposing structure on $\Theta_{\text{good}}$ and $\Theta_{\text{bad}}$ the contamination model can capture interference among auction markets. We assume that submarkets are separated, which models the ideal case where there is no interference. A buyer $i \in C_k$ is only interested in items from the submarket he belongs to: $v_i(\theta) = 0$ for $\theta \in \Theta_{k'}, k' \neq k$. Next, we let $\Theta_{\text{bad}}$ represent items that cause outbound edges from submarkets; see the green edges in Fig 2 left panel. An item is referred to as *bad* if it has positive values for buyers from at least two different submarkets. Formally, $\theta \in \Theta_{\text{bad}}$ if there exist $i \in C_k$, $j \in C_{k'}$, $k \neq k'$, such that $v_i(\theta) > 0$ and $v_j(\theta) > 0$. Combining these assumptions, we have that a buyer $i$ from submarket $k$ has positive values *only* for items from the sets $\Theta_k$ and possibly $\Theta_{\text{bad}}$. Now we have fully specified a contaminated market setup: we wish to make inferences on the market consisting of only $\Theta_{\text{good}}$ (which is really $K$ fully separate submarkets), but we observe an actual market containing items from $\Theta_{\text{good}} \cup \Theta_{\text{bad}}$. With this setup, we can use the results developed in the following section to model interference in parallel submarkets.

In Fig 2 we present an example of interference among $K = 3$ submarkets. The market of interest is the perfectly separated market (right). This is because, in parallel A/B testing, submarkets are explicitly created such that each submarket resembles the global market. Then when a submarket receives a treatment, the observed quantities in that submarket, such as revenues and social welfare, are considered surrogates for the treatment effect in the global market. However, in practice we only observe the interfered finite market (left), which converges to the interfered limit market (middle). In App. F we show how to analyze parallel A/B testing using this framework.

## 4 A Debiased Estimator and Its Properties

This section develops a methodology for making inferences about the uncontaminated limit FPPE. Since the interference setting is a special case of the contamination setting, we develop theories for the latter. We introduce a surrogate for pacing multipliers, based on the notion of directional derivatives, and establish its debiasing property in Sec. 4.1. Then, we focus on estimating this surrogate quantity in Sec. 4.2, and develop asymptotic normality results in Sec. 4.3. Secondly, we consider estimating revenue, which can be thought of as a smooth function of pacing multipliers. We discuss debiased revenue estimation and inference based on our pacing multiplier results in App. E.

## 4.1 A Debiased Surrogate for Pacing Multipliers

If we view $\beta_\alpha^*$ as a function of the level of contamination $\alpha$, then one can imagine that under sufficient regularity conditions, the pacing multipliers in the perfectly separated market, $\beta_0^*$, can be approximated by some form of Taylor expansion of $\alpha \mapsto \beta_\alpha^*$ at $\alpha$. This can be made rigorous by the

notion of directional derivatives. We define

$$\mathrm{d}\boldsymbol{\beta}^*(\alpha) = \lim_{\epsilon \to 0^+} \frac{\boldsymbol{\beta}^*_{\alpha-\epsilon} - \boldsymbol{\beta}^*_\alpha}{\epsilon} . \tag{5}$$

if the limit exists. We will show in Thm. 1 that $\boldsymbol{\beta}^*_\alpha + \alpha \, \mathrm{d}\boldsymbol{\beta}^*(\alpha)$ serves as a good approximation to $\boldsymbol{\beta}^* = \boldsymbol{\beta}^*_0$.

Thanks to the convex program characterization of FPPE, the directional derivative $\mathrm{d}\boldsymbol{\beta}^*(\alpha)$ has a closed-form expression under certain regularity conditions (the conditions are given in Thm. 1; the full proof is given in the appendix). We need a few notations for this expression. Define $\boldsymbol{\delta}_\alpha = \int \nabla f(\theta, \boldsymbol{\beta}^*_\alpha)(s - s') \, \mathrm{d}\theta$. Let $\boldsymbol{H}_\alpha = \nabla^2_{\boldsymbol{\beta}\boldsymbol{\beta}} \int F(\theta, \boldsymbol{\beta}^*_\alpha) s_\alpha \, \mathrm{d}\theta$ be the Hessian matrix in the market $\mathcal{M}_\alpha$ and $\boldsymbol{P}_\alpha = \mathrm{Diag}(1(\boldsymbol{\beta}^*_{\alpha,i} < 1))$. Then, under the regularity conditions given in Thm. 1 below,

$$\mathrm{d}\boldsymbol{\beta}^*(\alpha) = -(\boldsymbol{P}_\alpha \boldsymbol{H}_\alpha \boldsymbol{P}_\alpha)^\dagger \boldsymbol{\delta}_\alpha . \tag{6}$$

We present a heuristic derivation in App. G. Given the closed-form expression of $\mathrm{d}\boldsymbol{\beta}^*(\alpha)$, we define the following debiased pacing multiplier

$$\widetilde{\boldsymbol{\beta}}^* = \boldsymbol{\beta}^*_\alpha + \alpha \cdot (-(\boldsymbol{P}_\alpha \boldsymbol{H}_\alpha \boldsymbol{P}_\alpha)^\dagger \boldsymbol{\delta}_\alpha) . \tag{7}$$

**Theorem 1** (Analysis of Bias). *Suppose that in the market $\mathcal{M}_0$ conditions* SMO *and* SCS *hold, and assume that $\boldsymbol{\beta} \mapsto \nabla^2 \int F(\theta, \boldsymbol{\beta}) s' \, \mathrm{d}\theta$ is twice continuously differentiable in a neighborhood of $\boldsymbol{\beta}^*$. Then the directional derivative $\mathrm{d}\boldsymbol{\beta}^*(\cdot)$ is well-defined in a neighborhood of zero, and is given by Eq. (6). Moreover, as $\alpha \downarrow 0$,*

$$\|\widetilde{\boldsymbol{\beta}}^* - \boldsymbol{\beta}^*\|_2 = o(\alpha) .$$

The proof is given in App. B.1. Thm. 1 indicates that the debiased surrogate $\widetilde{\boldsymbol{\beta}}^*$ removes first-order bias caused by contamination. The limit pacing multipliers $\boldsymbol{\beta}^*_\alpha$ of the contaminated market $\mathcal{M}_\alpha$ will have bias of order $\boldsymbol{\beta}^*_\alpha - \boldsymbol{\beta}^* = \Theta(\alpha)$. In contrast, Thm. 1 shows that the debiased surrogate only incurs a bias of order $o(\alpha)$.

## 4.2 THE ESTIMATOR

In this section we introduce a plug-in estimator for the debiased surrogate $\widetilde{\boldsymbol{\beta}}^*$ and introduce a consistency theorem. The next section discusses constructing confidence intervals.

To estimate $\mathrm{d}\boldsymbol{\beta}^*(\alpha)$ in Eq. (6) we need estimates of its three components: the Hessian $\boldsymbol{H}_\alpha = \nabla^2_{\boldsymbol{\beta}\boldsymbol{\beta}} \int F(\theta, \boldsymbol{\beta}^*_\alpha) s_\alpha \, \mathrm{d}\theta$, the diagonal matrix $\boldsymbol{P}_\alpha$ and the vector $\boldsymbol{\delta}_\alpha = \int \nabla f(\theta, \boldsymbol{\beta}^*_\alpha)(s - s') \, \mathrm{d}\theta$.

**The Hessian**. For simplicity in our theoretical results, we will simply assume a generic Hessian estimator $\widehat{\boldsymbol{H}}_\alpha$ such that for some $\eta_t \downarrow 0$ we have $\widehat{\boldsymbol{H}}_\alpha - \boldsymbol{H}_\alpha = O_p(\eta_t)$. Hong et al. (2015) discuss the estimation of the derivative in detail. Different kinds of statistical guarantees require different rate conditions on $\eta_t$; see Theorems 2 and 3. We then introduce two Hessian estimators: one is applicable for general FPPE, while the other requires an extra market regularity condition. The first Hessian estimator is the finite difference method. Let $\boldsymbol{e}_i, \boldsymbol{e}_j$ be basis vectors and $\varepsilon_t$ be a step-size. Then the estimator is

$$\widehat{\boldsymbol{H}}_\alpha[i,j] = [H_{\alpha,t}(\boldsymbol{\beta}^\gamma_{++}) - H_{\alpha,t}(\boldsymbol{\beta}^\gamma_{+-}) - H_{\alpha,t}(\boldsymbol{\beta}^\gamma_{-+}) + H_{\alpha,t}(\boldsymbol{\beta}^\gamma_{--})]/(4\varepsilon_t^2) ,$$

where $\boldsymbol{\beta}^\gamma_{\pm\pm} = \boldsymbol{\beta}^\gamma_\alpha \pm \boldsymbol{e}_i \varepsilon_t \pm \boldsymbol{e}_j \varepsilon_t$, and $H_{\alpha,t}(\boldsymbol{\beta}) = \frac{1}{t} \sum_{\tau=1}^t F(\theta^\tau, \boldsymbol{\beta})$, with $\{\theta^\tau\}_\tau$ being the items in $\widehat{\mathcal{M}}_\alpha$. In practice, a diagonal approximation of the Hessian suffices. The second method relies on an additional regularity condition, in which case we derive a simplified formula for the Hessian, thereby enabling a simpler estimation procedure (see Thm. 3).

**The vector $\boldsymbol{\delta}_\alpha$.** Let $g$ be the Radon-Nikodym ratio $g(\theta) = (\mathrm{d}(s - s')/\mathrm{d}s_\alpha)(\theta) = \frac{1}{1-\alpha} 1_{\Theta_{\text{good}}}(\theta) - \frac{1}{\alpha} 1_{\Theta_{\text{bad}}}(\theta)$. With the ratio $g$, the true vector $\boldsymbol{\delta}_\alpha$ can be written as $\boldsymbol{\delta}_\alpha = \int g(\theta) \nabla f(\theta, \boldsymbol{\beta}^*_\alpha) s_\alpha(\theta) \, \mathrm{d}\theta$, which is easy to estimate given i.i.d. draws from $s_\alpha$. In particular, our estimator is then $\widehat{\boldsymbol{\delta}}_\alpha = \frac{1}{t} \sum_{\tau=1}^t g(\theta^\tau) \boldsymbol{\mu}^\tau$ . Here $\boldsymbol{\mu}^\tau = [x_1^\tau v_1^\tau, \ldots, x_n^\tau v_n^\tau]^\intercal$ is a subgradient of $f(\theta^\tau, \boldsymbol{\beta}^\gamma_\alpha)$ w.r.t. $\boldsymbol{\beta}$.

**The diagonal matrix $\boldsymbol{P}_\alpha$.** Recall $\boldsymbol{P}_\alpha = \mathrm{Diag}(1(\boldsymbol{\beta}^*_{\alpha,i} < 1))$. So a natural estimator is $\widehat{\boldsymbol{P}}_\alpha = \mathrm{Diag}(1(\boldsymbol{\beta}^\gamma_{\alpha,i} < 1 - \iota_t))$, where the slackness $\iota_t \asymp \frac{1}{\sqrt{t}}$.

With all three components estimated, using $\beta_\alpha^\gamma$ is the pacing multiplier in the market $\widehat{\mathscr{M}}_\alpha$ we define the plug-in estimator for $\mathrm{d}\beta^*(\alpha)$ in Eq. (6) as

$$\widehat{\beta} = \beta_\alpha^\gamma - \alpha \cdot (\widehat{P}_\alpha \widehat{H}_\alpha \widehat{P}_\alpha)^\dagger \widehat{\delta}_\alpha \ . \tag{8}$$

**Theorem 2** (Consistency). *Suppose that in the market $\mathscr{M}_\alpha$ conditions* SMO *and* SCS *hold. If the Hessian estimation error satisfies $\eta_t = o(1)$, then $\widehat{\beta} \xrightarrow{p} \widetilde{\beta}^*$. The proof is in App. B.2.*

### 4.3 ASYMPTOTIC NORMALITY AND INFERENCE

We present two asymptotic normality results. In the first result, we require a stronger condition on the Hessian error rate $\eta_t$. In particular, as will be shown in Thm. 3, the rate condition $\eta_t = o(1/\sqrt{t})$ is sufficient for normality. One could use a separate large historical dataset to obtain a good estimate of the Hessian matrix. In the second result, we impose an additional condition on market structure which simplifies the Hessian expression and facilitates efficient Hessian estimation. To describe the additional market structure, we define the gap between the highest and the second-highest bid for an item $\theta$ under pacing $\beta$ by $\mathsf{bidgap}(\beta, \theta) = \max\{\beta_i v_i(\theta)\} - \mathrm{secondmax}\{\beta_i v_i(\theta)\}$ , where $\mathrm{secondmax}$ is the second-highest entry potentially equal to the highest; e.g., $\mathrm{secondmax}([1, 1, 2]) = 1$. When there is a tie for an item $\theta$ under pacing $\beta$, we have $\mathsf{bidgap}(\beta, \theta) = 0$. When there is no tie for an item $\theta$, the gap $\mathsf{bidgap}(\beta, \theta)$ is strictly positive.

For any $g : \Theta \to \mathbb{R}$, let $\mathbb{E}_\alpha[g] = \int g s_\alpha \,\mathrm{d}\theta$ and $Cov_\alpha(g) = \mathbb{E}_\alpha[(g - \mathbb{E}_\alpha[g])(g - \mathbb{E}_\alpha[g])^\intercal]$. Recall $\widetilde{\beta}^*$ is the debiased surrogate in Eq. (7) and $\widehat{\beta}$ is its estimator defined in Eq. (8).

We need to introduce a few more notations to describe the normality results. First, let $d_\alpha = -(P_\alpha H_\alpha P_\alpha)^\dagger \nabla f(\cdot, \beta_\alpha^*)$ . As mentioned previously, the pacing multipliers in the contaminated market converge to the limit counterpart and have the representation

$$\sqrt{t}(\beta_\alpha^\gamma - \beta_\alpha^*) = \frac{1}{\sqrt{t}} \sum_{\tau=1}^t (d_\alpha(\theta^\tau) - \mathbb{E}_\alpha[d_\alpha(\theta^\tau)]) + o_p(1) \ .$$

In the statistics literature, the function $d_\alpha(\cdot) - \mathbb{E}[d_\alpha]$ is called the influence function (van der Vaart, 2000). For our debiased estimators, we need the following (uncentered) influence functions.

$$d_1(\theta) = \left(\tfrac{1}{1-\alpha} 1_{\Theta_{\mathrm{good}}}(\theta)\right) d_\alpha(\theta) \ , \ \ d_2(\theta) = d_1(\theta) - 2\alpha \mathrm{Diag}(\beta_{\alpha,i}^* \delta_{\alpha,i} b_i^{-1}) d_\alpha(\theta) \ .$$

**Theorem 3.** *Let* SCS *and* SMO *hold in $\mathscr{M}_\alpha$.*

*1. **Asymptotic Normality in a General Market.** It holds that $\widehat{\beta} - \widetilde{\beta}^* = z_t + O_p(\eta_t) + o_p(\tfrac{1}{\sqrt{t}})$, where $\sqrt{t} z_t \xrightarrow{d} \mathcal{N}(0, \Sigma_1)$ with $\Sigma_1 = Cov_\alpha(d_1)$, and $\eta_t$ is the Hessian estimation error.*

*2. **Asymptotic Normality under a Bid Gap Condition.** If in addition $\mathbb{E}_\alpha[1/\mathsf{bidgap}(\beta_\alpha^*, \theta)] < \infty$, then $H_\alpha = \mathrm{Diag}(b_i/(\beta_{\alpha,i}^*)^2)$. Suppose we estimate $H_\alpha$ with $\widehat{H}_\alpha = \mathrm{Diag}(b_i/(\beta_{\alpha,i}^\gamma)^2)$. Then $\sqrt{t}(\widehat{\beta} - \widetilde{\beta}^*) \xrightarrow{d} \mathcal{N}(0, \Sigma_2)$ where $\Sigma_2 = Cov_\alpha(d_2)$. The proof is in App. B.3.*

Thm. 3 part 1 shows how the error of the Hessian estimate affects the distribution of the estimator $\widehat{\beta}$. If $\eta_t = o(1/\sqrt{t})$, then the decomposition becomes $\sqrt{t}(\widehat{\beta} - \widetilde{\beta}^*) = \sqrt{t} z_t + o_p(1)$, implying asymptotic normality, i.e. $\sqrt{t}(\widehat{\beta} - \widetilde{\beta}^*) \xrightarrow{d} \mathcal{N}(0, \Sigma_1)$, in which case one can construct an ellipsoidal confidence region for $\widetilde{\beta}^*$. Thm. 3 part 2 shows direct asymptotic normality under the extra condition, with a simpler Hessian estimator that avoids finite differences.

To perform inference, we need to construct a consistent estimate of the covariance matrix. Now we describe a plug-in estimate of $\Sigma_1$. Let the estimator $\widehat{d_1}^\tau$ be $\widehat{d_1}^\tau = -\left(\tfrac{1}{1-\alpha} 1_{\Theta_{\mathrm{good}}}(\theta^\tau)\right)(\widehat{P}_\alpha \widehat{H}_\alpha \widehat{P}_\alpha)^\dagger \mu^\tau$ , where $\mu^\tau = [x_1^\tau v_1^\tau, \ldots, x_n^\tau v_n^\tau]^\intercal$, and $\widehat{P}_\alpha, \widehat{H}_\alpha$ have been defined in Sec. 4.2. The plug-in estimator is $\widehat{\Sigma}_1 = \tfrac{1}{t} \sum_{\tau=1}^t (\widehat{d_1}^\tau - \overline{d_1})(\widehat{d_1}^\tau - \overline{d_1})^\intercal$ with $\overline{d_1} = \tfrac{1}{t} \sum_{\tau=1}^t \widehat{d_1}^\tau$. By similar arguments as in Liao et al. (2023), the plug-in estimates of $\Sigma_1$ and $\Sigma_2$ are consistent. Algorithm 1 summarizes the debiasing procedure.

In App. E we present a similar debiased estimator for revenue and its bias and variance properties. In App. F we specialize the debiased estimator to parallel budget-controlled A/B testing.

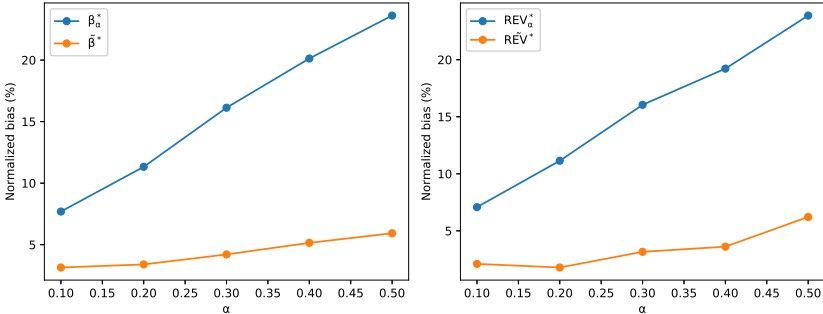

**Figure 3:** Normalized bias (in percent of true value) as a function of $\alpha$ in semi-synthetic experiments. $\widetilde{\boldsymbol{\beta}}^*$ and $\widetilde{REV}^*$ are the debiased surrogates for pacing multiplier and revenue in the limit market with interference $\mathscr{M}_\alpha$.

## 5 SEMI-SYNTHETIC EXPERIMENT

To evaluate our proposed framework and debiased estimator, we run semi-synthetic simulations to check if the proposed estimator for beta and revenue are indeed less biased, and we test the coverage of the proposed estimator. Fully synthetic experiments are presented in App. H.

In the semi-synthetic experiments, we simulate 40 buyers and 10000 *good* items in two submarkets, with a varying number of *bad* items (up to 5000) in order to study the effect of the contamination parameter $\alpha$. For each $\alpha$, we randomly sample a budget for each buyer, and compute $\boldsymbol{\beta}^*$ and $REV^*$ from the limit pure market $\mathscr{M}_0$ with a value function in each submarket. Both the budget and values are sampled from historical bidding data, making the budget and value distributions heavy-tailed as in the real-world applications. More specifically, we first sample a certain number of auctions. For each auction, we sample a given number of advertisers with their per-impression bids. Advertisers that are sampled across different auctions are treated as the same buyers and their budgets are determined by aggregating their values over auctions up to a scalar to calibrated to get the percentage of budget-constrained buyers equal to what was observed in the real-world auction market, along the same lines as the experiments of Conitzer et al. (2022b).

To check if the debiased surrogate reduces bias, we compute $\boldsymbol{\beta}_\alpha^*$ and $REV_\alpha^*$ from the limit market with interference $\mathscr{M}_\alpha$ and their surrogates, $\widetilde{\boldsymbol{\beta}}^*$ and $\widetilde{REV}^*$ ($\widetilde{\boldsymbol{\beta}}^*$ is defined Eq. (7), $\widetilde{REV}^*$ is defined in App. E. We look at the normalized bias for the surrogate, defined as $\|\widetilde{\boldsymbol{\beta}}^* - \boldsymbol{\beta}^*\|_2 / \|\boldsymbol{\beta}^*\|_2$ for pacing multipliers and as $|\widetilde{REV}^* / REV^* - 1|$ for revenue, and similarly defined for the limit quantities. Fig 3 shows the normalized bias curves as a function of $\alpha$. The magnitude of the bias increases with $\alpha$, for both the variables in the limit market with interference $\mathscr{M}_\alpha$ and their debiased surrogates. The bias of the debiased surrogates is indeed much smaller than the contaminated limit quantities.

Next, we check the coverage of the proposed variance estimator. For each $\alpha$ and each budget sample, we run 100 simulations in the following way: We sample items (or their values for each buyer) considering two submarkets and bad items. We then run the finite FPPE with bad items and obtain a baseline estimate for pacing multiplier and revenue without applying the debiasing procedure. Then, we apply the debiasing procedure to compute the debiased estimates. For each simulation, we check if the debiased surrogate is within the confidence interval of the debiased estimator. Finally, we aggregate them to compute the estimated coverage of the estimator. The results for both pacing multiplier and revenue are shown in Table 1. For the coverage of $\widehat{\boldsymbol{\beta}}$, we first compute the coverage of each component and report only the average in the table. For revenue, we construct the CI using the two approaches as mentioned in App. E: one based on Eq. (22) and the other using parametric bootstrap based on the estimated asymptotic distribution of $\widehat{\boldsymbol{\beta}}$ (with "(b)" in the column names).

Firstly, Fig 4 shows that both CIs converges to the true value from the limit market with interfrence $\mathscr{M}_\alpha$ as the number of items goes to infinity. Then, in Table 1, we show that the coverage for $\widehat{\boldsymbol{\beta}}$ is slightly smaller than the nominal level (95%), as well as the coverage of the bootstrap CI of

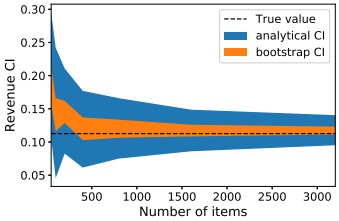

**Figure 4:** Revenue confidence intervals as a function of the number of items in semi-synthetic experiments. The analytic CI comes from Eq. (22). The true value is the debiased surrogates for revenue in the limit market with interference $\mathcal{M}_\alpha$.

revenue. The under-coverage for $\widehat{\boldsymbol{\beta}}$ is mainly driven by the under-estimation of the variance of $\widehat{\boldsymbol{\beta}}$, while the under-coverage of the bootstrap CI for revenue can also be partially attributed to the higher dimensionality (with 40 buyers), making the bootstrap resampling harder to explore the whole space.

Although the proposed variance estimator has good asymptotic properties, the results from our synthetic experiments suggest that it can perform badly, in either direction, for finite markets. Constructing more accurate variance estimators for our debiased estimator in finite settings would certainly mitigate the over- or under-coverage issues that we observe here and deserve more future research.

One promising alternative is to construct the CI for $\widehat{\boldsymbol{\beta}}$ and $\widehat{REV}$ by directly bootstrapping the observed value matrix, though this might work best for independent valuations across buyers.

| | $\widehat{\boldsymbol{\beta}}$ | $\widehat{REV}$ | | | |
|---|---|---|---|---|---|
| $\alpha$ | coverage | CI width | CI width (b) | coverage | coverage (b) |
| 1000/11000 | 0.877 | 0.244 | 0.044 | 1.0 | 0.95 |
| 2000/12000 | 0.849 | 0.225 | 0.043 | 1.0 | 0.87 |
| 3000/13000 | 0.852 | 0.210 | 0.041 | 1.0 | 0.81 |
| 4000/14000 | 0.828 | 0.200 | 0.040 | 1.0 | 0.90 |
| 5000/15000 | 0.826 | 0.191 | 0.039 | 1.0 | 0.90 |

**Table 1:** Coverage of $\widehat{\boldsymbol{\beta}}$ and revenue estimates in the semi-synthetic experiments. All quantities are averaged over 100 simulations for each $\alpha$ (the ratio of the number of bad items and the total items). The coverage of $\widehat{\boldsymbol{\beta}}$ is averaged over all components of $\widehat{\boldsymbol{\beta}}$. For revenue estimates, the columns with "(b)" represent the quantities from the bootstrap CI, while the columns without "(b)" are for the CI from Eq. (22). The CI widths are normalized by the revenue from the limit market $\mathcal{M}_\alpha$.

## 6   CONCLUSION

We have proposed a practical experimental design for performing concurrent A/B tests in large-scale ad auction markets, using a submarket clustering approach, and showed that in production experiments, this submarket clustering approach leads to strong sign consistency performance, as compared to A/B testing on the full market, while allowing significantly-higher A/B test throughput. In order to model the potential for interference between submarket A/B tests, we introduced a theoretical model of statistical inference in first-price pacing equilibrium problems, under settings with supply contamination. We showed how one can perform statistical inference in such a setting using a debiased estimation procedure, and studied the statistical properties of this procedure. We then showed how our model of statistical inference in FPPE with contamination can be used to model the submarket clustering parallel A/B test design, and gave theoretical performance guarantees. Finally, we presented numerical experiments on fully synthetic and semi-synthetic data derived from Meta ad auctions. The experiments showed that our proposed debiased estimator achieves smaller biases and its statistical coverage on realistic data is generally in line with the predictions from our theory.

ACKNOWLEDGEMENTS

We would like to thank the anonymous reviewers for their useful comments. Christian Kroer and Lu-ofeng Liao were supported by the Office of Naval Research awards N00014-22-1-2530 and N00014-23-1-2374, and the National Science Foundation awards IIS-2147361 and IIS-2238960.

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

## A    Technical Lemmas

### A.1    A perturbation result for constrained stochastic programs

We introduce a theorem from Shapiro (1990). Theorem 1 in that paper handles the case where the constraints are also defined as expectations of some random functions, and constraints are also perturbed in the analysis. For simplicity we specialize the theorem to deterministic constraints and do not perturb constraints. Note that Theorem 7.27 from Shapiro et al. (2021) can also be used to prove our Thm. 1.

Let $\Theta$ be a probability space equipped with an appropriate $\sigma$-algebra. Consider $F : \Theta \times \mathbb{R}^n \to \mathbb{R}$ and a set $B \subset \mathbb{R}^n$, given by

$$B = \{\boldsymbol{\beta} : g_i(\boldsymbol{\beta}) = 0, i \in I; g_i(\boldsymbol{\beta}) \leq 0, i \in J\}.$$

where $I$ and $J$ are finite index sets. Let $P$ and $Q$ be two probability measures on $\Theta$. Let $\phi(\boldsymbol{\beta}, \alpha) = (P + \alpha(Q - P))F(\cdot, \boldsymbol{\beta})$ and $\phi(\boldsymbol{\beta}) = \phi(\boldsymbol{\beta}, 0)$. Here $\min_B \phi$ is the main program of interest and $Q$ is a perturbation measure. The amount of perturbation is measured by $\alpha \in [0, 1]$.

Let $\boldsymbol{\beta}^*$ be the unique minimizer of $\phi(\boldsymbol{\beta})$ over $B$. Let $J^* \subset J$ be the inequality constraints active at $\boldsymbol{\beta}^*$, meaning $g_i(\boldsymbol{\beta}^*) = 0$ for all $i \in J^*$. Let $\boldsymbol{\beta}^*(\alpha)$ be the unique optimal solution to $\min_B \phi(\cdot, \alpha)$.

**Lemma 1** (Theorem 1 from Shapiro (1990)). *Define the Lagrangian function by $L(\boldsymbol{\beta}, \lambda, \alpha) = \phi(\boldsymbol{\beta}, \alpha) + \sum_{i \in I \cup J} \lambda_i g_i(\boldsymbol{\beta})$ and $L_0(\boldsymbol{\beta}, \lambda) = L(\boldsymbol{\beta}, \lambda, 0)$. Let $\Lambda_0$ be the set of optimal Lagrangian multipliers [1]. Define the critical cone $C = \{\boldsymbol{u} : \boldsymbol{u}^\intercal \nabla g_i(\boldsymbol{\beta}^*) = 0, i \in I; \boldsymbol{u}^\intercal \nabla g_i(\boldsymbol{\beta}^*) \leq 0$ for $i \in J^*; \boldsymbol{u}^\intercal \nabla \phi(\boldsymbol{\beta}^*) \leq 0\}$. Define $\Lambda_0^* = \arg\max_{\lambda \in \Lambda_0} \sum_{i \in I \cup J} \lambda_i g_i(\boldsymbol{\beta}^*)$. Assume the following conditions.*

1. *Differentiability. The functions $\phi(\cdot, 0)$, $\phi(\cdot, 1)$, and $g_i, i \in I \cup J$, are continuously differentiable in a neighborhood of $\boldsymbol{\beta}^*$.*

2. *Constraint Qualification. The gradients $\nabla g_i(\boldsymbol{\beta}^*)$, $i \in I$ are linearly independent. And there exists $b \in \mathbb{R}^n$ such that $b^\intercal \nabla g_i(\boldsymbol{\beta}^*) = 0$ for $i \in I$ and $b^\intercal \nabla g_i(\boldsymbol{\beta}^*) < 0$ for $i \in J^*$.*

3. *Differentiability. The functions $\phi(\cdot, 0)$, $g_i, i \in I \cup J^*$ are twice continuously differentiable in a neighbourhood of $\boldsymbol{\beta}^*$.*

4. *Second-order sufficient condition. Assume for all nonzero $\boldsymbol{u} \in C$, $\max_{\lambda \in \Lambda_0^*} \boldsymbol{u}^\intercal \nabla_{xx}^2 L_0(\boldsymbol{\beta}^*, \lambda)\boldsymbol{u} > 0$.*

*Define the function $b$ and the set $\Sigma$ by*

$$b(\boldsymbol{u}) = \max_{\lambda \in \Lambda_0^*} \boldsymbol{u}^\intercal \nabla_{xt}^2 L_0(\boldsymbol{\beta}^*, \lambda) + \frac{1}{2}\boldsymbol{u}^\intercal \nabla_{xx}^2 L_0(\boldsymbol{\beta}^*, \lambda)\boldsymbol{u} \tag{9}$$

$$\Sigma = \arg\min \boldsymbol{u}^\intercal \nabla \phi(\boldsymbol{\beta}^*) \tag{10}$$
$$\text{s.t.} \quad \boldsymbol{u}^\intercal \nabla g_i(\boldsymbol{\beta}^*) = 0, i \in I; \boldsymbol{u}^\intercal \nabla g_i(\boldsymbol{\beta}^*) \leq 0, i \in J^*;$$

*Then (i) there exists a positive $K$ such that $\|\boldsymbol{\beta}^*(\alpha) - \boldsymbol{\beta}^*\| \leq K\alpha$ for all positive $\alpha$ in a neighborhood of zero. (ii) If, in addition, the function $b$ has a unique minimizer $\boldsymbol{u}$ over $\Sigma$, then the limit $\lim_{\alpha \downarrow 0}(\boldsymbol{\beta}^*(\alpha) - \boldsymbol{\beta}^*)/\alpha$ exists and equals to $\boldsymbol{u}$.*

## B PROOFS

### B.1 PROOF OF CLOSED-FORM EXPRESSION FOR BIAS AND THM. 1

Now we prove Thm. 1.

To begin with, we define directional derivatives. For a probability measure $P$ on the item set $\Theta$, let $\boldsymbol{\beta}(P)$ be the unique optimal solution to the Eisenberg-Gale program: $\boldsymbol{\beta}(P) = \min_{\boldsymbol{\beta} \in (0,1]^n} \int F(\theta, \boldsymbol{\beta}) \, dP(\theta)$. Let $P$ and $Q$ be two measures. When exists, the directional derivative is defined as

$$d\boldsymbol{\beta}(P; Q - P) = \lim_{t \downarrow 0} \frac{\boldsymbol{\beta}(P + t(Q - P)) - \boldsymbol{\beta}(P)}{t}.$$

**Step 1.** Show there exist $K > 0$ and $\bar{\alpha}_1 > 0$ such that $\|\boldsymbol{\beta}_\alpha^* - \boldsymbol{\beta}^*\| \leq K\alpha$ for $\alpha \in [0, \bar{\alpha}_1]$. And $d\boldsymbol{\beta}(s; s - s') = -(\boldsymbol{P}\boldsymbol{H}_0\boldsymbol{P})^\dagger \boldsymbol{\delta}_0$.

We apply lemma 1 with $F(\theta, \boldsymbol{\beta}) = \max_i v_i(\theta)\beta_i - \sum_i b_i \log(\beta_i)$, $g_i(\boldsymbol{\beta}) = \beta_i - 1$, $B = (0, 1]^n$, $dP = s \, d\theta$, $d(Q - P) = (s - s') \, d\theta$.

By SCS in the market $\mathcal{M}_0$, the set $\Sigma$ (defined in Eq. (10)) becomes the plane $\{\boldsymbol{u} : u_i = 0, i \in I_+\}$ where we recall $I_+ = \{i : \beta_i^* = 0\}$. By SMO in $\mathcal{M}_0$, we know the Lagrangian multiplier of the EG program in $\mathcal{M}_0$ is unique, and thus $b(\cdot)$ defined in Eq. (9) becomes $b(\boldsymbol{u}) = \boldsymbol{u}^\intercal \boldsymbol{\delta}_0 + \frac{1}{2}\boldsymbol{u}^\intercal \boldsymbol{H}_0 \boldsymbol{u}$. We conclude the directional derivative is $d\boldsymbol{\beta}(s; s - s') = \lim_{\epsilon \to 0^+}(\boldsymbol{\beta}_0^* - \boldsymbol{\beta}_{-\epsilon}^*)/\epsilon = -(\boldsymbol{P}\boldsymbol{H}_0\boldsymbol{P})^\dagger \boldsymbol{\delta}_0$, where $\boldsymbol{P} = \text{Diag}(1(\beta_i^* < 1))$ and $\boldsymbol{\delta}_0 = \int \nabla f(\cdot, \boldsymbol{\beta}^*)(s - s') \, d\theta$

**Step 2.** For all $\alpha \geq 0$ small enough, it holds $I_\alpha = I$, where we recall $I_\alpha = \{i : \beta_{\alpha,i}^* = 1\}$ and $I = \{i : \beta_i^* = 0\}$.

First we show $I \subseteq I_\alpha$, i.e., $\{i : \beta_i^* = 1\} \subseteq \{i : \beta_{\alpha,i}^* = 1\}$ for $\alpha$ small enough. This is saying, if a constraint $\beta_i \leq 1$ is strongly active in the EG program of $\mathcal{M}_0$, then it is also strongly active

---

[1] $\lambda \in \Lambda_0$ iff $\nabla_x L_0(\beta_0, \lambda) = 0$ and $\lambda_i \geq 0, i \in J^*$ and $\lambda_i = 0$ for all $j \in J \setminus J^*$.

in the EG program of the market $\mathcal{M}_\alpha$. This holds by Lemma 2.2 from Shapiro (1988) and the SCS condition in $\mathcal{M}_0$. Next, we show, for small $\alpha$, $I_\alpha \subseteq I$ by showing $[n] \setminus I \subseteq [n] \setminus I_\alpha$, i.e., $\{i : \beta_i^* < 1\} \subseteq \{i : \beta_{\alpha,i}^* < 1\}$. This holds by $\|\boldsymbol{\beta}_\alpha^* - \boldsymbol{\beta}^*\| \leq K\alpha$ for all $\alpha \leq \bar{\alpha}_1$.

**Step 3.** For all $\alpha \geq 0$ small enough, the directional derivative $\mathrm{d}\boldsymbol{\beta}(s_\alpha; s - s') = \lim_{\epsilon \to 0^+} (\boldsymbol{\beta}_{\alpha-\epsilon}^* - \boldsymbol{\beta}_\alpha^*)/\epsilon$ exists and $\alpha \mapsto \mathrm{d}\boldsymbol{\beta}(s_\alpha; s - s')$ is continuous.

Applying lemma 1 with $\mathrm{d}P = s_\alpha \, \mathrm{d}\theta$ and $\mathrm{d}(Q - P) = (s - s') \, \mathrm{d}\theta$ shows that the directional derivative exists for all $\alpha$ small enough. Now we find an expression for $\mathrm{d}\boldsymbol{\beta}(s_\alpha; s - s')$. By SMO in $\mathcal{M}_0$, twice continuously differentiability of $\boldsymbol{\beta} \mapsto \nabla^2 \int F(\boldsymbol{\beta}, \theta) s' \, \mathrm{d}\theta$ in a neighborhood of $\boldsymbol{\beta}^*$, and the Lipschitzness result from step 1, we know $\boldsymbol{\beta} \to \nabla^2 \int F(\theta, \boldsymbol{\beta}) s_\alpha \, \mathrm{d}\theta$ is twice continuously differentiable at $\boldsymbol{\beta}_\alpha^*$, which implies uniqueness of Lagrangian multiplier in the market $\mathcal{M}_\alpha$, for all $\alpha$ small enough (Lemma 2.2 from Shapiro (1988)). And thus $b(\cdot)$ defined in Eq. (9) becomes $b(\boldsymbol{u}) = \boldsymbol{u}^\intercal \boldsymbol{\delta}_\alpha + \frac{1}{2} \boldsymbol{u}^\intercal \boldsymbol{H}_\alpha \boldsymbol{u}$. Next, the SCS condition in the market $\mathcal{M}_0$ implies SCS in $\mathcal{M}_\alpha$ for all $\alpha$ small enough. The set $\Sigma$ (defined in Eq. (10)) becomes the plane $\{\boldsymbol{u} : u_i = 0, i \in I_\alpha\}$. The quadratic program $\min_\Sigma b$ has a closed-form solution: $\boldsymbol{u} = -(\boldsymbol{P}_\alpha \boldsymbol{H}_\alpha \boldsymbol{P}_\alpha)^\dagger \boldsymbol{\delta}_\alpha$ where $\boldsymbol{P}_\alpha = \mathrm{Diag}(1(\beta_{\alpha,i}^* < 1))$, $\boldsymbol{\delta}_\alpha = \int \nabla f(\cdot, \boldsymbol{\beta}_\alpha^*)(s - s') \, \mathrm{d}\theta$. And we apply $I_\alpha = I$ for all $\alpha$ small enough from step 2, and so $\boldsymbol{P}_\alpha = \boldsymbol{P}$. We conclude

$$\mathrm{d}\boldsymbol{\beta}(s_\alpha; s - s') = -(\boldsymbol{P} \boldsymbol{H}_\alpha \boldsymbol{P})^\dagger \boldsymbol{\delta}_\alpha$$

for all $\alpha$ small enough.

Next, we show that the map $\alpha \mapsto (\boldsymbol{P} \boldsymbol{H}_\alpha \boldsymbol{P})^\dagger \boldsymbol{\delta}_\alpha$ is continuous for all $\alpha$ small enough. To see this, note $\boldsymbol{H}_\alpha = \nabla^2 \int F(\theta, \boldsymbol{\beta}_\alpha^*) s_\alpha \, \mathrm{d}\theta = (1-\alpha) \nabla^2 \int F(\cdot, \boldsymbol{\beta}_\alpha^*) s \, \mathrm{d}\theta + \alpha \nabla^2 \int F(\cdot, \boldsymbol{\beta}_\alpha^*) s' \, \mathrm{d}\theta$ is continuous in $\alpha$. The $\boldsymbol{P}$ matrix is fixed, and so $\alpha \mapsto \boldsymbol{P} \boldsymbol{H}_\alpha \boldsymbol{P}$ is continuous. Without loss of generality, suppose $I^c = \{i : \beta_i^* < 0\} = [k]$ are the first $k$ buyers. Then $\boldsymbol{P} \boldsymbol{H}_\alpha \boldsymbol{P}$ creates a matrix with upper left $k$-by-$k$ block equal to the upper left $k$-by-$k$ block of $\boldsymbol{H}_\alpha$, denoted $\boldsymbol{H}_{\alpha, I^c I^c}$, and zeros everywhere else. Then $(\boldsymbol{P} \boldsymbol{H}_\alpha \boldsymbol{P})^\dagger$ is a matrix with upper left block being $(\boldsymbol{H}_{\alpha, I^c I^c})^{-1}$ and zeros everywhere else. Since $\boldsymbol{H}_\alpha$ is positive definite for all small $\alpha$, the submatrix $\boldsymbol{H}_{\alpha, I^c I^c}$ must also be positive definite. And so $\alpha \mapsto (\boldsymbol{H}_{\alpha, I^c I^c})^{-1}$ is continuous, implying continuity of $\alpha \mapsto (\boldsymbol{P} \boldsymbol{H}_\alpha \boldsymbol{P})^\dagger$. Finally, $\boldsymbol{\delta}_\alpha = \int \nabla f(\cdot, \boldsymbol{\beta}_\alpha^*)(s - s') \, \mathrm{d}\theta$ is continuous in $\alpha$.

**Step 4.** The desired claim: $\widetilde{\boldsymbol{\beta}}^* - \boldsymbol{\beta}^* = o(\alpha)$. Note $\mathrm{d}\boldsymbol{\beta}^*(\alpha)$ defined in Eq. (6) is exactly $\mathrm{d}\boldsymbol{\beta}(s_\alpha; s - s')$. And so

$$
\begin{aligned}
\widetilde{\boldsymbol{\beta}}^* - \boldsymbol{\beta}^* &= \boldsymbol{\beta}_\alpha^* + \alpha \, \mathrm{d}\boldsymbol{\beta}(s_\alpha; s - s') - \boldsymbol{\beta}^* \\
&= (\boldsymbol{\beta}^* - \alpha \, \mathrm{d}\boldsymbol{\beta}(s_0; s - s') + o(\alpha)) + \alpha \, \mathrm{d}\boldsymbol{\beta}(s_\alpha; s - s') - \boldsymbol{\beta}^* \\
&= \alpha(\mathrm{d}\boldsymbol{\beta}(s_\alpha; s - s') - \mathrm{d}\boldsymbol{\beta}(s_0; s - s')) + o(\alpha) = o(\alpha).
\end{aligned}
$$

where the last line uses continuity in $\alpha$ of the directional derivative. This completes the proof of Thm. 1.

### B.2 PROOF OF THM. 2

Recall the estimator $\widehat{\boldsymbol{\beta}} = \boldsymbol{\beta}_\alpha^\gamma - \alpha(\widehat{\boldsymbol{P}}_\alpha \widehat{\boldsymbol{H}}_\alpha \widehat{\boldsymbol{P}}_\alpha)^\dagger \widehat{\boldsymbol{\delta}}_\alpha$ and the debiased surrogate is $\widetilde{\boldsymbol{\beta}}^* = \boldsymbol{\beta}_\alpha^* - \alpha(\boldsymbol{P}_\alpha \boldsymbol{H}_\alpha \boldsymbol{P}_\alpha)^\dagger \boldsymbol{\delta}_\alpha$. By results from Liao & Kroer (2023) we know that SCS and SMO in the market $\mathcal{M}_\alpha$ and $\eta_t = o(1)$ imply $\boldsymbol{\beta}_\alpha^\gamma \xrightarrow{p} \boldsymbol{\beta}_\alpha^*$ and $(\widehat{\boldsymbol{P}}_\alpha \widehat{\boldsymbol{H}}_\alpha \widehat{\boldsymbol{P}}_\alpha)^\dagger \xrightarrow{p} (\boldsymbol{P}_\alpha \boldsymbol{H}_\alpha \boldsymbol{P}_\alpha)^\dagger$. Finally, $\widehat{\boldsymbol{\delta}}_\alpha \xrightarrow{p} \boldsymbol{\delta}_\alpha$ holds by the law of large numbers. We complete the proof of Thm. 2.

### B.3 PROOF OF THM. 3

Recall the estimator $\widehat{\boldsymbol{\beta}} = \boldsymbol{\beta}_\alpha^\gamma - \alpha(\widehat{\boldsymbol{P}}_\alpha \widehat{\boldsymbol{H}}_\alpha \widehat{\boldsymbol{P}}_\alpha)^\dagger \widehat{\boldsymbol{\delta}}_\alpha$ and the debiased surrogate is $\widetilde{\boldsymbol{\beta}}^* = \boldsymbol{\beta}_\alpha^* - \alpha(\boldsymbol{P}_\alpha \boldsymbol{H}_\alpha \boldsymbol{P}_\alpha)^\dagger \boldsymbol{\delta}_\alpha$. Define

$$\boldsymbol{z}_t = \boldsymbol{\beta}_\alpha^\gamma - \boldsymbol{\beta}_\alpha^* - \alpha(\boldsymbol{P}_\alpha \boldsymbol{H}_\alpha \boldsymbol{P}_\alpha)^\dagger(\widehat{\boldsymbol{\delta}}_\alpha - \boldsymbol{\delta}_\alpha) \tag{11}$$

$$\boldsymbol{\xi}_t = -\alpha((\widehat{\boldsymbol{P}}_\alpha \widehat{\boldsymbol{H}}_\alpha \widehat{\boldsymbol{P}}_\alpha)^\dagger - (\boldsymbol{P}_\alpha \boldsymbol{H}_\alpha \boldsymbol{P}_\alpha)^\dagger) \boldsymbol{\delta}_\alpha \tag{12}$$

$$\boldsymbol{\zeta}_t = -\alpha((\widehat{\boldsymbol{P}}_\alpha \widehat{\boldsymbol{H}}_\alpha \widehat{\boldsymbol{P}}_\alpha)^\dagger - (\boldsymbol{P}_\alpha \boldsymbol{H}_\alpha \boldsymbol{P}_\alpha)^\dagger)(\widehat{\boldsymbol{\delta}}_\alpha - \boldsymbol{\delta}_\alpha) \tag{13}$$

Then clearly $\widehat{\boldsymbol{\beta}} - \widetilde{\boldsymbol{\beta}}^* = \boldsymbol{z}_t + \boldsymbol{\xi}_t + \boldsymbol{\zeta}_t$.

Step 1. We show $\sqrt{t}z_t$ converges to a normal distribution. By Liao & Kroer (2023), it holds

$$\sqrt{t}(\boldsymbol{\beta}_\alpha^\gamma - \boldsymbol{\beta}_\alpha^*) = \frac{1}{\sqrt{t}}\sum_{\tau=1}^{t} \boldsymbol{d}_\alpha(\theta^\tau) - \mathbb{E}[\boldsymbol{d}_\alpha] + o_p(1) \tag{14}$$

where $\boldsymbol{d}_\alpha(\theta) = -(\boldsymbol{P}_\alpha \boldsymbol{H}_\alpha \boldsymbol{P}_\alpha)^\dagger \nabla f(\theta, \boldsymbol{\beta}_\alpha^*)$ is the influence function. Next, we define the likelihood ratio function $g(\theta) = (\mathrm{d}(s - s')/\mathrm{d}s_\alpha)(\theta) = -\frac{1}{\alpha}1_{\Theta_{\text{bad}}}(\theta) + \frac{1}{1-\alpha}1_{\Theta_{\text{good}}}(\theta)$. Then $\boldsymbol{\delta}_\alpha = \int \nabla f(\cdot, \boldsymbol{\beta}_\alpha^*)(s - s') \, \mathrm{d}\theta = \int g(\cdot)\nabla f(\cdot, \boldsymbol{\beta}_\alpha^*)s_\alpha(\cdot) \, \mathrm{d}\theta$. Then by definition $\widehat{\boldsymbol{\delta}}_\alpha$ can be written as $\widehat{\boldsymbol{\delta}}_\alpha = P_t(g(\cdot)\nabla f(\cdot, \boldsymbol{\beta}_\alpha^\gamma))$ where $P_t h(\cdot) = \frac{1}{t}\sum_{\tau=1}^{t} h(\theta^\tau)$ and $\theta^\tau$ are i.i.d. draws from $s_\alpha$. Also let $\nu_t h = \sqrt{t}(P_t h - \int h s_\alpha \, \mathrm{d}\theta)$. By the decomposition,

$$\sqrt{t}(\widehat{\boldsymbol{\delta}}_\alpha - \boldsymbol{\delta}_\alpha) \tag{15}$$

$$= \sqrt{t}\left(P_t\big(g(\cdot)\nabla f(\cdot, \boldsymbol{\beta}_\alpha^\gamma)\big) - \int g(\cdot)\nabla f(\cdot, \boldsymbol{\beta}_\alpha^*)s_\alpha \, \mathrm{d}\theta\right) \tag{16}$$

$$= \sqrt{t}\left(P_t\big(g(\cdot)\nabla f(\cdot, \boldsymbol{\beta}_\alpha^*)\big) - \int g(\cdot)\nabla f(\cdot, \boldsymbol{\beta}_\alpha^*)s_\alpha \, \mathrm{d}\theta\right) + \nu_t\big(g(\cdot)(\nabla f(\cdot, \boldsymbol{\beta}_\alpha^\gamma) - \nabla f(\cdot, \boldsymbol{\beta}_\alpha^*))\big) \tag{17}$$

$$= \nu_t(g(\cdot)\nabla f(\cdot, \boldsymbol{\beta}_\alpha^*)) + o_p(1), \tag{18}$$

where the last line follows by a stochastic equicontinuity argument as in Liao & Kroer (2023). Finally, $\boldsymbol{d}_\alpha(\cdot) - \alpha g(\cdot)(\boldsymbol{P}_\alpha \boldsymbol{H}_\alpha \boldsymbol{P}_\alpha)^\dagger \nabla f(\cdot, \boldsymbol{\beta}_\alpha^*)$ is exactly the influence function $\boldsymbol{d}_1(\cdot)$ defined in the theorem.

Step 2. Show $\boldsymbol{\xi}_t = O_p(\eta_t)$.

We need a matrix perturbation result.

**Lemma 2** (Theorem 2.2 from Stewart (1977)). *Let $\boldsymbol{A}$ and $\boldsymbol{B} = \boldsymbol{A} + \boldsymbol{E}$ be nonsingular square matrices, and $\|\boldsymbol{A}^{-1}\|\|\boldsymbol{E}\| < 1$. Here $\|\cdot\|$ is the operator norm. Then*

$$\|\boldsymbol{B}^{-1} - \boldsymbol{A}^{-1}\| \leq \frac{\|\boldsymbol{A}^{-1}\|^2}{1 - \|\boldsymbol{E}\|\|\boldsymbol{A}^{-1}\|}\|\boldsymbol{E}\|. \tag{19}$$

By Liao & Kroer (2023), we know $\mathbb{P}(\widehat{\boldsymbol{P}}_\alpha = \boldsymbol{P}_\alpha) \to 1$. Without loss of generality, suppose $I_\alpha^c = \{i : \beta_{\alpha,i}^* < 0\} = [k]$ are the first $k$ buyers. Then $(\boldsymbol{P}_\alpha \boldsymbol{H}_\alpha \boldsymbol{P}_\alpha)$ creates a matrix with upper left $k$-by-$k$ block equal to the upper left $k$-by-$k$ block of $\boldsymbol{H}_\alpha$, denoted $\boldsymbol{H}_{\alpha,I^cI^c}$, and zeros everywhere else. Under the event $\{\widehat{\boldsymbol{P}}_\alpha = \boldsymbol{P}_\alpha\}$, the matrix $(\widehat{\boldsymbol{P}}_\alpha \widehat{\boldsymbol{H}}_\alpha \widehat{\boldsymbol{P}}_\alpha)$ is one with upper left $k$-by-$k$ block being upper left $k$-by-$k$ block of $\widehat{\boldsymbol{H}}_\alpha$, denoted $\widehat{\boldsymbol{H}}_{\alpha,I^cI^c}$, and zeros everywhere else. Now let $\boldsymbol{A} = \widehat{\boldsymbol{H}}_{\alpha,I^cI^c}$ and $\boldsymbol{B} = \boldsymbol{H}_{\alpha,I^cI^c}$. It is clear that $\|\boldsymbol{\xi}_t\|_2 \leq \alpha\|\boldsymbol{\delta}_\alpha\|_2\|\boldsymbol{A}^{-1} - \boldsymbol{B}^{-1}\| = O_p(\eta_t)$ by lemma 2.

Step 3. Show $\boldsymbol{\zeta}_t = o_p(\frac{1}{\sqrt{t}})$. From previous derivation, $\widehat{\boldsymbol{\delta}}_\alpha - \boldsymbol{\delta}_\alpha = O_p(1/\sqrt{t})$ and $(\widehat{\boldsymbol{P}}_\alpha \widehat{\boldsymbol{H}}_\alpha \widehat{\boldsymbol{P}}_\alpha)^\dagger - (\boldsymbol{P}_\alpha \boldsymbol{H}_\alpha \boldsymbol{P}_\alpha)^\dagger = O_p(\eta_t) = o_p(1)$. We conclude $\boldsymbol{\zeta}_t = O_p(1/\sqrt{t})o_p(1) = o_p(1/\sqrt{t})$.

We complete the proof of Thm. 3 part 1.

Now we prove part 2. The claim that $\boldsymbol{H}_\alpha = \mathrm{Diag}(b_i/(\beta_{\alpha,i}^*)^2)$ follows from Liao et al. (2023). Next we derive the asymptotic distribution. We only need to handle the $\boldsymbol{\xi}_t$ term defined as before. Under the event $\{\widehat{\boldsymbol{P}}_\alpha = \boldsymbol{P}_\alpha\}$, we have $\boldsymbol{\xi}_t = -\mathrm{Diag}(\alpha\delta_{\alpha,i}1(\beta_{\alpha,i}^* < 1)b_i^{-1})(\boldsymbol{\beta}_\alpha^\gamma \circ \boldsymbol{\beta}_\alpha^\gamma - \boldsymbol{\beta}_\alpha^* \circ \boldsymbol{\beta}_\alpha^*)$. By Liao & Kroer (2023) we know $\sqrt{t}(\boldsymbol{\beta}_\alpha^\gamma - \boldsymbol{\beta}_\alpha^*) = \frac{1}{\sqrt{t}}\sum_{\tau=1}^{t}(\boldsymbol{d}_\alpha(\theta^\tau) - \mathbb{E}_\alpha[\boldsymbol{d}_\alpha]) + o_p(1)$. So by the delta method,

$$\sqrt{t}(\boldsymbol{\beta}_\alpha^\gamma \circ \boldsymbol{\beta}_\alpha^\gamma - \boldsymbol{\beta}_\alpha^* \circ \boldsymbol{\beta}_\alpha^*) = \mathrm{Diag}(2\boldsymbol{\beta}_\alpha^*)\frac{1}{\sqrt{t}}\sum_{\tau=1}^{t}(\boldsymbol{d}_\alpha(\theta^\tau) - \mathbb{E}_\alpha[\boldsymbol{d}_\alpha]) + o_p(1).$$

Summarizing,

$$\sqrt{t}\boldsymbol{\xi}_t = -\mathrm{Diag}(2\beta_{\alpha,i}^*\alpha\delta_{\alpha,i}1(\beta_{\alpha,i}^* < 1)b_i^{-1})\frac{1}{\sqrt{t}}\sum_{\tau=1}^{t}(\boldsymbol{d}_\alpha(\theta^\tau) - \mathbb{E}_\alpha[\boldsymbol{d}_\alpha]) + o_p(1)$$

Note that for $i$ such that $\beta^*_{\alpha,i} = 1$, the $i$-th entry of $\boldsymbol{d}_\alpha$ will be zero almost surely, so the indicator can be dropped in the diagonal matrix. Now let $\boldsymbol{d}_2(\theta) = -\mathrm{Diag}(2\alpha\beta^*_{\alpha,i}\delta_{\alpha,i}b_i^{-1})\boldsymbol{d}_\alpha + \boldsymbol{d}_1$, where we recall $\boldsymbol{d}_1$ is defined in Thm. 3. Then under the condition that $\mathbb{E}_\alpha[\frac{1}{\mathsf{bidgap}(\boldsymbol{\beta}^*_\alpha,\theta)}] < \infty$, we have

$$\sqrt{t}(\widehat{\boldsymbol{\beta}} - \widetilde{\boldsymbol{\beta}}^*) \xrightarrow{d} \mathscr{N}(0, \mathbb{E}_\alpha[(\boldsymbol{d}_2 - \mathbb{E}_\alpha[\boldsymbol{d}_2])(\boldsymbol{d}_2 - \mathbb{E}_\alpha[\boldsymbol{d}_2])^\intercal]).$$

### B.4 Proof of Thm. 4

*Proof.* Note

$$\left| \int \max_i(v_i(\theta)\widetilde{\boldsymbol{\beta}}^*_i)s(\theta)\,\mathrm{d}\theta - \int \max_i(v_i(\theta)\boldsymbol{\beta}^*_i)s(\theta)\,\mathrm{d}\theta \right|$$
$$\leq \int |\max_i(v_i(\theta)\widetilde{\boldsymbol{\beta}}^*_i)s(\theta) - \max_i(v_i(\theta)\boldsymbol{\beta}^*_i)s(\theta)|\,\mathrm{d}\theta$$
$$\leq \max_i \sup_\theta v_i(\theta) \cdot \|\boldsymbol{\beta}^* - \widetilde{\boldsymbol{\beta}}^*\|_\infty = o(\alpha)$$

as $\alpha \downarrow 0$.

Let $P_t h(\cdot) = \frac{1}{t}\sum_{\tau=1}^t h(\theta^\tau)$ and $\theta^\tau$ are i.i.d. draws from $s_\alpha$, and $\boldsymbol{P}h(\cdot) = \int h s_\alpha$. Let $g(\cdot) = \mathrm{d}s/\mathrm{d}s_\alpha = \frac{1}{1-\alpha}\mathbb{1}_{\Theta_{\mathsf{good}}}$. Also let $\nu_t h = \sqrt{t}(P_t h - \int h s_\alpha\,\mathrm{d}\theta)$. We change the measure and write $\widetilde{\widehat{REV}}^* = \int f(\theta, \widetilde{\boldsymbol{\beta}}^*)s\,\mathrm{d}\theta = \int f(\theta, \widetilde{\boldsymbol{\beta}}^*)g(\theta)s_\alpha\,\mathrm{d}\theta$. For the claim regarding asymptotic normality, note

$$\sqrt{t}(\widehat{REV} - \widetilde{\widehat{REV}}^*)$$
$$= \sqrt{t}[P_t f(\cdot, \widehat{\boldsymbol{\beta}})g(\cdot) - Pf(\cdot, \widetilde{\boldsymbol{\beta}}^*)g(\cdot)]$$
$$= \sqrt{t}[P_t f(\cdot, \widetilde{\boldsymbol{\beta}}^*)g(\cdot) - Pf(\cdot, \widetilde{\boldsymbol{\beta}}^*)g(\cdot)] + \sqrt{t}[Pf(\cdot, \widehat{\boldsymbol{\beta}})g(\cdot) - Pf(\cdot, \widetilde{\boldsymbol{\beta}}^*)g(\cdot)]$$
$$\quad + \sqrt{t}(P_t - P)(f(\cdot, \widehat{\boldsymbol{\beta}})g(\cdot) - f(\cdot, \widetilde{\boldsymbol{\beta}}^*)g(\cdot))$$
$$= \nu_t(f(\cdot, \widetilde{\boldsymbol{\beta}}^*)g(\cdot)) + \nu_t(\mathbb{E}_\alpha[g(\cdot)\nabla f(\cdot, \widetilde{\boldsymbol{\beta}}^*)]^\intercal \boldsymbol{d}(\cdot)) + o_p(1)$$

where the last line follows by the delta method and a stochastic equicontinuity argument. We complete the proof of the theorem. □

## C Ommited Maintext

### C.1 Formal Definition of Finite FPPE

**Definition 5** (Finite FPPE, Conitzer et al. (2022a))**.** *Given $(b, v, \sigma, \gamma)$, the finite FPPE, $\widehat{FPPE}(\boldsymbol{b}, \boldsymbol{v}, \sigma, \gamma)$, is the unique tuple $(\boldsymbol{\beta}, p) \in [0,1]^n \times \mathbb{R}_+^t$ such that there exist $x_i^\tau \in [0,1]$: (First-price) For all $\tau$, $p^\tau = \max_i \beta_i v_i^\tau$. For all $i$ and $\tau$, $x_i^\tau > 0$ implies $\beta_i v_i^\tau = \max_k \beta_k v_k^\tau$. (Supply and budget feasible) For all $i$, $\sigma\sum_{\tau=1}^t x_i^\tau p^\tau \leq b_i$. For all $\tau$, $\sum_{i=1}^n x_i^\tau \leq 1$. (Market clearing) For all $\tau$, $p^\tau > 0$ implies $\sum_{i=1}^n x_i^\tau = 1$. (No unnecessary pacing) For all $i$, $\sigma\sum_{\tau=1}^t x_i^\tau p^\tau < b_i$ implies $\beta_i = 1$.*

## D Related Work

**Pacing equilibrium.** Pacing is a budget management strategy where bids are scaled down by a multiplicative factor in order to smooth out and control spending. In the first-price setting, Borgs et al. (2007) study first price auctions with budget constraints in a perturbed model, where the limit prices converge to those of an FPPE. Building on that work, Conitzer et al. (2022a) introduce the FPPE framework to model autobidding in repeated auctions, and discover several properties of FPPE such as shill-proofness monotonicity properties, and a close relationship between FPPE and the quasilinear Fisher market equilibrium (Chen et al., 2007; Cole et al., 2017). Gao & Kroer (2022) propose an infinite-dimensional variant of the quasilinear Fisher market, which lays the probability foundations of the current paper. Gao et al. (2021) and Liao et al. (2022) study online computation of the

infinite-dimensional Fisher market equilibrium, and we utilize their methods to compute equilibria. In the second-price setting, a variety of models have been proposed for modeling the outcome of budget management: Balseiro et al. (2015) study budget-management in second-price auctions through a fluid mean-field approximation; Balseiro & Gur (2019) investigate adaptive pacing strategy from buyers' perspective in a stochastic continuous setting; and Balseiro et al. (2021b) study several budget smoothing methods including multiplicative pacing in a stochastic context. A second-price analogue of FPPE was explored by Conitzer et al. (2022b). Modeling statistical inference in the second-price setting is most likely harder, due to equilibrium multiplicity issues (Conitzer et al., 2022b), and hardness of even computing equilibria (Chen et al., 2023).

Recently, a series of papers (e.g., Aggarwal et al. (2019); Deng et al. (2021); Balseiro et al. (2021a)) investigated an "autobidding" model for value-maximizing bidders, who aim to maximize their value subject to budget or return-on-spend constraints, which may include buying individual ad impressions that have negative utility. Several extensions provide results for more general settings, such as the quality of equilibria for generalized rationality models for bidders Babaioff et al. (2021), non-uniform bidding strategies Deng et al. (2023), and liquid welfare guarantees when budget-constrained buyers fail to reach equilibrium Gaitonde et al. (2023). Statistical inference for parallel A/B testing under such models represent an interesting future direction of work.

**Interference in A/B Testing for Auction Markets.** Experimentation under interference has been extensively studied in the past decade within the context of social networks Ugander et al. (2013); Aronow & Samii (2017); Eckles et al. (2017); Athey et al. (2018); Li & Wager (2022) and two-sided marketplaces Zigler & Papadogeorgou (2018); Wager & Xu (2021). Under interference, the outcome of an experiment unit, either a user in a social network or a buyer in an online marketplace, can be affected by the treatment status of other units in the networks, violating the Stable Unit Treatment Values Assumption (SUTVA) assumption commonly employed in analyzing online A/B tests Rosenbaum (2007). To take into account this type of interference, novel designs and analysis approaches have been proposed, e.g., cluster randomization Ugander et al. (2013); Karrer et al. (2021), regression-based estimators Chin (2019), and bipartite analysis Harshaw et al. (2021). A/B testing in the presence of seller-side budget constraints has been discussed by Basse et al. (2016) and Liu et al. (2021) where the budget-splitting design spans the full market. However, most of the above works do not consider the particular type of interference caused by competitive equilibrium effects. Recent work has also addressed equilibrium effects (Wager & Xu, 2021; Liao et al., 2023; Liao & Kroer, 2024). Most relevant to our results, Liao & Kroer (2023) consider A/B testing via budget splitting in Fisher markets and FPPE. However, none of these works consider the effects caused by parallel A/B tests that interfere with each other. Our method builds on top of the budget-splitting design; our approach additionally segments the full market into submarkets, where different submarkets receive different treatment. In the formal sections of our paper we assume that the submarket structure is known. In practice, the submarkets may not be known; Rolnick et al. (2019) and Viviano et al. (2023) present methods to identify submarkets through clustering approaches that aim to limit the interference between clusters.

# E    RESULTS FOR REVENUE

---

**Algorithm 1:** Debiasing procedure

---

**Input.** Budgets $[b_1, \ldots, b_n]$. Values $v_i^\tau$. First-price allocation $x_i^\tau \in [0, 1]$. The observed equilibrium pacing multipliers $\boldsymbol{\beta}_\alpha^\gamma = [\boldsymbol{\beta}_1^\gamma, \ldots, \boldsymbol{\beta}_n^\gamma]$. Proportion of bad items $\alpha$. Desired confidence level $(1 - c) \in (0, 1)$.

1. Estimate the Hessian matrix $\widehat{\boldsymbol{H}}_\alpha$, e.g. via $\widehat{\boldsymbol{H}}_\alpha = \mathrm{Diag}(b_i/(\beta_{\alpha,i}^\gamma)^2)$, or using finite differences.
2. Compute relevant quantities $\widehat{\boldsymbol{P}}_\alpha, \widehat{\boldsymbol{\delta}}_\alpha, \boldsymbol{\mu}^\tau$.
3. Compute the debiased estimator $\widehat{\boldsymbol{\beta}}$ and covariance matrix $\widehat{\boldsymbol{\Sigma}}_1$.
4. Output confidence region: $\widehat{\boldsymbol{\beta}} + (\chi_{n,c}/\sqrt{t})\widehat{\boldsymbol{\Sigma}}_1^{1/2}\mathbb{B}$, where $\chi_{n,c}^2$ is the $(1 - c)$-quantile of a chi-square distribution with degree $n$, and $\mathbb{B}$ is the unit ball in $\mathbb{R}^n$.

---

The revenue in the good market is defined as $REV^* = \int \max_i\{v_i(\theta)\beta_i^*\}s(\theta)\,\mathrm{d}\theta$. We define the debiased revenue surrogate and its estimator

$$\widetilde{REV}^* = \int \max_i\{v_i(\theta)\widetilde{\beta}_i^*\}s(\theta)\,\mathrm{d}\theta \ , \tag{20}$$

$$\widehat{REV} = \frac{1}{1-\alpha}\frac{1}{t}\sum_{\tau=1}^{t}1_{\Theta_{\mathsf{good}}}(\theta^\tau)\max_i\{v_i(\theta^\tau)\widehat{\beta}_i\} \ . \tag{21}$$

In words, we approximate $REV^*$ with the revenue generated in the market where buyers bid according to the pacing profile $\widetilde{\beta}^*$, and then we estimate this via plug-in estimation on the subset of items that are good.

**Theorem 4.** *Let the conditions from Thm. 1 and Thm. 3 hold. The debiased revenue removes first-order bias: $\widetilde{REV}^* = REV^* + o(\alpha)$. Assume that $\sqrt{t}(\widehat{\beta} - \widetilde{\beta}^*)$ has influence function $d$, i.e., $\sqrt{t}(\widehat{\beta} - \widetilde{\beta}^*) = \frac{1}{\sqrt{t}}\sum_{\tau=1}^{t}d(\theta^\tau) + o_p(1)$, then $\sqrt{t}(\widehat{REV} - \widetilde{REV}^*) \xrightarrow{d} \mathcal{N}(0, \sigma_{REV}^2)$ where $\sigma_{REV}^2 = Cov_\alpha(d_{REV})$, $d_{REV}(\cdot) = \frac{1}{1-\alpha}\big(1_{\Theta_{\mathsf{good}}}(\cdot)f(\cdot, \widetilde{\beta}^*) + \mathbb{E}_\alpha[\nabla f(\theta, \widetilde{\beta}^*)1_{\Theta_{\mathsf{good}}}(\theta)]^\mathsf{T}d(\cdot)\big)$. The proof is in App. B.4.*

Examples of influence functions compatible with Thm. 4 are the influence functions in Thm. 3.

We have two methods to construct CIs for the debiased revenue surrogate in Eq. (7). The first method is based on the asymptotic normality results in Thm. 4. In the second method, a revenue CI is constructed based on a CI for $\beta_\alpha^*$. Suppose $\mathbb{CI}$ is some confidence region for $\beta_\alpha^*$. Then $\{\frac{1}{1-\alpha}\frac{1}{t}\sum_{\tau=1}^{t}1_{\Theta_{\mathsf{good}}}(\theta^\tau)\max_i v_i(\theta^\tau)\beta_i : \beta \in \mathbb{CI}\}$ serves as a CI for revenue. Another way to utilize a pacing multiplier CI is based on the observation that revenue is a monotone function of the pacing multiplier. Let $\underline{\beta}_i = \min\{\beta_i : \beta \in \mathbb{CI}\}$ and $\bar{\beta}_i = \max\{\beta_i : \beta \in \mathbb{CI}\}$. Then another natural (potentially wider) CI for $\widetilde{REV}^*$ is $[\underline{REV}, \overline{REV}]$, where

$$\underline{REV} = \frac{1}{t}\sum_{\tau=1}^{t}\frac{1_{\Theta_{\mathsf{good}}}(\theta^\tau)}{1-\alpha}\max_i v_i(\theta^\tau)\underline{\beta}_i \ , \quad \overline{REV} = \frac{1}{t}\sum_{\tau=1}^{t}\frac{1_{\Theta_{\mathsf{good}}}(\theta^\tau)}{1-\alpha}\max_i v_i(\theta^\tau)\bar{\beta}_i \ . \tag{22}$$

## F  PARALLEL A/B TEST UNDER INTERFERENCE

In this section, we use the theory from the previous sections to formulate a statistical inference theory for parallel A/B tests under interference.

Consider $K$ auction markets in which we want to run parallel A/B tests. There are $n$ buyers, each with budget $b_i$ corresponding to one unit supply of items, and an item set $\Theta$, partitioned into $\Theta_{\mathsf{good}}$ and $\Theta_{\mathsf{bad}}$. Let $C_1, \ldots, C_K$ be a partition of the buyers, $\Theta_1, \ldots, \Theta_K$ be a partition of the good item set $\Theta_{\mathsf{good}}$, and $s_1, \ldots s_K$ be a set of supply functions, supported on $\Theta_1, \ldots, \Theta_K$ respectively. The $k$-th submarket consists of buyers in $C_k$, the item set $\Theta_k$ and supply $s_k$. Let $s = \frac{1}{K}\sum_k s_k$ be the average mixture and $s'$ be a supply supported on $\Theta_{\mathsf{bad}}$.

To model treatment application we introduce the *potential value functions*

$$v(0) = (v_1(0, \cdot), \ldots, v_n(0, \cdot)), \ v(1) = (v_1(1, \cdot), \ldots, v_n(1, \cdot)) \ .$$

If item $\theta$ is exposed to treatment $w \in \{0, 1\}$, then its value to buyer $i$ will be $v_i(w, \theta)$.

We extend the structure on $\Theta_{\mathsf{bad}}$ and $\Theta_{\mathsf{good}}$ introduced in Sec. 3.2 to A/B testing here. We assume these submarkets are separated. This is to model the ideal case where there is no interference. A buyer $i \in C_k$ is only interested in items from the submarket he belongs to: $v_i(w, \theta) = 0$ for $\theta \in C_{k'}, k' \neq k$, and $w \in \{0, 1\}$. An item is called a bad item if it has positive values for buyers from different submarkets. Formally, $\theta \in \Theta_{\mathsf{bad}}$ if there exist $w \in \{0, 1\}, i \in C_k, j \in C_{k'}, k \neq k'$, so that $v_i(w, \theta) > 0$ and $v_j(w, \theta) > 0$. A buyer $i$ from submarket $k$ has a (potentially) positive value only for items in $\Theta_k \cup \Theta_{\mathsf{bad}}$.

**Step 1.  The treatment effects**. Let $\mathscr{M}(w, k) = \mathsf{FPPE}(\{b_i\}_{i \in C_k}, \{v_i(\cdot, w)\}_{i \in C_k}, s_k, \Theta_k)$ be the $k$-th limit submarket under treatment $w \in \{0, 1\}$, and let revenue be $REV_k^*(w)$. Then the treatment

effect in the submarket $k$ is $\tau_k^* = REV_k^*(1) - REV_k^*(0)$. Equivalently, all $K$ submarkets can be formed simultaneously, $\mathscr{M}(w) = \text{FPPE}\left(\frac{1}{K}\boldsymbol{b}, \boldsymbol{v}(w), s, \Theta\right)$. The scaling $1/K$ in budgets ensures that a buyer in $C_k$ has budget $b_i/K$ to bid for items in $\Theta_k$, whose supply in $\mathscr{M}(w)$ is $\frac{1}{K}s_k$. The $k$-th component of $\mathscr{M}(w)$ corresponds to $\mathscr{M}(w, k)$.

**Step 2. The experiment.** A practical parallel A/B test scheme is as follows.

*Step 2.1. Budget splitting.* Decide on a budget split ratio $\pi_0, \pi_1$ satisfying $\pi_0 + \pi_1 = 1$, and split budgets accordingly. Choose A/B test size $t$, the total number of impressions in each submarket summed across treatments. Then, the budget of buyer $i$ for treatment 0 will be $\pi_0 t b_i$, and $\pi_1 t b_i$ for treatment 1. This is because $b_i$ is normalized to correspond to one unit of total supply.

*Step 2.2. Observe markets with interference.* Draw items/impressions from $\alpha s' + (1-\alpha)s$. Draw $\lfloor \pi_0 t \rfloor$ items for each submarket under treatment 0 and $\lfloor \pi_1 t \rfloor$ for treatment 1. A total of $t_w = \lfloor \pi_w t K \rfloor$ items are drawn for the whole market under treatment $w$. Since items come from the contaminated distribution, roughly a small fraction $\alpha$ of the $t_w$ items cause interference.

The data observed following the described A/B test scheme can be compactly represented by two finite FPPEs (one for each treatment) (Def. 5): $\widehat{\mathscr{M}_\alpha}(w) = \widehat{\text{FPPE}}(\pi_w t \boldsymbol{b}, \boldsymbol{v}(w), 1, \gamma)$, where $\gamma \subset \Theta$ consists of $t_w = \lfloor \pi_w t K \rfloor$ i.i.d. draws from the mixture $\alpha s' + (1-\alpha)s$.

It can be shown that the observed market $\widehat{\mathscr{M}_\alpha}(w)$ converges to $\mathscr{M}_\alpha(w) = \text{FPPE}\left(\frac{1}{K}\boldsymbol{b}, \boldsymbol{v}(w), s_\alpha, \Theta\right)$.

**Step 3. Debiasing $\beta$.** Following previous section, we can debias $\widehat{\mathscr{M}_\alpha}(w)$ to approximate $\mathscr{M}(w)$. Let $\widehat{\boldsymbol{\beta}}(w)$ be the debiased pacing multiplier.

**Step 4. Revenue estimator.** Estimate $\widehat{REV}_k(w)$ by

$$\frac{1}{\pi_w(1-\alpha)t} \sum_{\tau=1}^{t_w} 1_{\Theta_k}(\theta^\tau) \max_{i \in C_k} v_i(\theta^\tau) \widehat{\beta}_i(w) .$$

Then $\widehat{\tau}_k = \widehat{REV}_k(1) - \widehat{REV}_k(0)$ and a variance estimate can be obtained by either asymptotic normality, confidence region of $\boldsymbol{\beta}$, or bootstrap.

## G   HEURISTIC DERIVATION OF THE DEBIASED SURROGATE

We present a heuristic argument for Eq. (6) here; for a rigorous treatment we need a perturbation theory result which is given in lemma 1 in the appendix. Define the contaminated EG objective $H_\alpha(\boldsymbol{\beta}) = \int F(\boldsymbol{\beta}, \theta) s_\alpha \, d\theta$. Then $\boldsymbol{H}_\alpha = \nabla^2 H_\alpha(\boldsymbol{\beta}_\alpha^*)$. The map $(\boldsymbol{\beta}, \alpha) \mapsto H_\alpha(\boldsymbol{\beta})$ has gradient $[\nabla H(\boldsymbol{\beta}^*), \int F(\boldsymbol{\beta}_\alpha^*, \theta)(s' - s) \, d\theta]^\top$ and Hessian $[\boldsymbol{H}_\alpha, -\boldsymbol{\delta}_\alpha; -\boldsymbol{\delta}_\alpha^\top, 0]$ at $(\boldsymbol{\beta}_\alpha^*, \alpha)$. Then we have the following quadratic approximation of $H_{\alpha-\epsilon}(\boldsymbol{\beta})$

$$H_{\alpha-\epsilon}(\boldsymbol{\beta}) \approx H_\alpha(\boldsymbol{\beta}_\alpha^*) + \begin{pmatrix} \boldsymbol{\beta} - \boldsymbol{\beta}_\alpha^* \\ -\epsilon \end{pmatrix}^\top \begin{pmatrix} \nabla H_\alpha(\boldsymbol{\beta}_\alpha^*) \\ \int F(\boldsymbol{\beta}_\alpha^*, \theta)(s' - s) \, d\theta \end{pmatrix}$$
$$+ \frac{1}{2} \begin{pmatrix} \boldsymbol{\beta} - \boldsymbol{\beta}_\alpha^* \\ -\epsilon \end{pmatrix}^\top \begin{pmatrix} \boldsymbol{H}_\alpha & -\boldsymbol{\delta}_\alpha \\ -\boldsymbol{\delta}_\alpha^\top & 0 \end{pmatrix} \begin{pmatrix} \boldsymbol{\beta} - \boldsymbol{\beta}_\alpha^* \\ -\epsilon \end{pmatrix} .$$

And so

$$\boldsymbol{\beta}_{\alpha-\epsilon}^* = \underset{\boldsymbol{\beta} \in (0,1]^n}{\arg\min} H_{\alpha-\epsilon}(\boldsymbol{\beta}) \approx \underset{\boldsymbol{\beta} \in (0,1]^n}{\arg\min} (\nabla H_\alpha(\boldsymbol{\beta}_\alpha^*) + \epsilon \boldsymbol{\delta}_\alpha)^\top \boldsymbol{\beta}$$
$$+ \frac{1}{2}(\boldsymbol{\beta} - \boldsymbol{\beta}_\alpha^*)^\top \boldsymbol{H}_\alpha (\boldsymbol{\beta} - \boldsymbol{\beta}_\alpha^*) ,$$

where we dropped terms that do not involve $\boldsymbol{\beta}$.

Next we need two observations. For $\epsilon$ small enough, we have $\beta_{\alpha-\epsilon,i}^* = 1$ if $\beta_{\alpha,i}^* = 1$. Intuitively, buyers in the market $\mathscr{M}_\alpha$ with $\beta_{\alpha,i}^* = 1$ have some nonzero leftover budget, which acts as a buffer to small changes in supply, and thus their pacing multipliers remain one. Next, we also note $\boldsymbol{\beta} \mapsto \nabla H_\alpha(\boldsymbol{\beta}_\alpha^*)^\top \boldsymbol{\beta}$ is constant on the set $\{\boldsymbol{\beta} : \beta_i = 1 \text{ if } \beta_{\alpha,i}^* = 1\}$ by complementary slackness. Then we

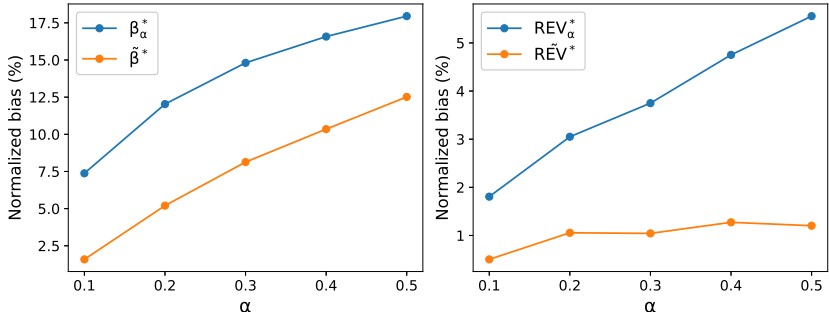

**Figure 5:** Normalized bias (in percent of true value) as a function of $\alpha$ in fully synthetic experiments. $\widetilde{\boldsymbol{\beta}}^*$ and $\widetilde{REV}^*$ are the debiased surrogates for pacing multiplier and revenue in the limit market with interference $\mathscr{M}_\alpha$.

further simplify and obtain

$$\boldsymbol{\beta}^*_{\alpha-\epsilon} \approx \underset{\beta_i=1 \text{ if } \beta^*_{\alpha,i}=1}{\arg\min} \, \epsilon\boldsymbol{\delta}^\top_\alpha\boldsymbol{\beta} + \tfrac{1}{2}(\boldsymbol{\beta} - \boldsymbol{\beta}^*_\alpha)^\top \boldsymbol{H}_\alpha(\boldsymbol{\beta} - \boldsymbol{\beta}^*_\alpha) \,.$$

The right-hand side is just a linearly constrained quadratic optimization problem, which admits a closed-form solution $\boldsymbol{\beta}^*_\alpha - \epsilon(\boldsymbol{P}_\alpha\boldsymbol{H}_\alpha\boldsymbol{P}_\alpha)^\dagger\boldsymbol{\delta}_\alpha$.

## H    FULLY SYNTHETIC EXPERIMENTS

In the fully synthetic experiments, we simulate 10 buyers and 1000 *good* items in two submarkets, with a varying number of *bad* items (up to 500) in order to study the effect of the contamination parameter $\alpha$. For each $\alpha$, we randomly sample a budget from a uniform distribution for each buyer, and compute $\boldsymbol{\beta}^*$ and $REV^*$ from the limit pure market $\mathscr{M}_0$ with a uniformly distributed value function in each submarket. Fig 5 shows the normalized bias curves as a function of $\alpha$ for the fully synthetic experiment, which is similar to that from semi-synthetic experiments in Fig 3.

We also check the coverage of the proposed variance estimator in fully synthetic experiments. The coverage of $\widehat{\boldsymbol{\beta}}$ and revenue are estimated in the same way as in the semi-synthetic ones. It is shown in Fig 6 that both CIs converges to the true value from the limit market with interfrence $\mathscr{M}_\alpha$ as the number of items goes to infinity. Then, in  Table 2, we show that the bootstrap CI width is much smaller compared to that from Eq. (22). However, unlike in the semi-synthetic experiments, both CIs are over-covered, higher than the nominal level (95%). It is expected though that the CI from Eq. (22) is relatively large and hence conservative since it uses the min and max of each $\beta_i$, hence likely living outside of the confidence region ($n$-dimension ellipsoid) for $\widehat{\boldsymbol{\beta}}$ and leading to over-coverage. For the bootstrap CI, it requires enough bootstrap samples to fully explore the whole boundary of the confidence region of $\widehat{\boldsymbol{\beta}}$, as well as an accurate estimate of the asymptotic distribution of $\widehat{\boldsymbol{\beta}}$. In our fully synthetic simulation, our hypothesis is that the over-coverage of the bootstrap CI is mainly due to the over-estimation of the variance for $\widehat{\boldsymbol{\beta}}$.

| | $\widehat{\boldsymbol{\beta}}$ | $\widehat{REV}$ | | | |
|---|---|---|---|---|---|
| $\alpha$ | coverage | CI width | CI width (b) | coverage | coverage (b) |
| 100/1100 | 0.993 | 0.132 | 0.043 | 1.0 | 1.0 |
| 200/1200 | 0.994 | 0.141 | 0.059 | 1.0 | 1.0 |
| 300/1300 | 0.980 | 0.107 | 0.059 | 1.0 | 1.0 |
| 400/1400 | 0.983 | 0.098 | 0.064 | 1.0 | 1.0 |
| 500/1500 | 0.971 | 0.108 | 0.069 | 1.0 | 1.0 |

**Table 2:** Coverage of $\widehat{\boldsymbol{\beta}}$ and revenue estimates in fully synthetic experiments. All quantities are averaged over 100 simulations for each $\alpha$. The coverage of $\widehat{\boldsymbol{\beta}}$ is averaged over all components of $\widehat{\boldsymbol{\beta}}$. For revenue estimates, the columns with "(b)" represent the quantities from the bootstrap CI, while the columns without "(b)" are for the CI from Eq. (22). The CI widths are normalized by the revenue from the limit market $\mathcal{M}_\alpha$. $\alpha$ is expressed as the ratio of the number of bad items and the total items.

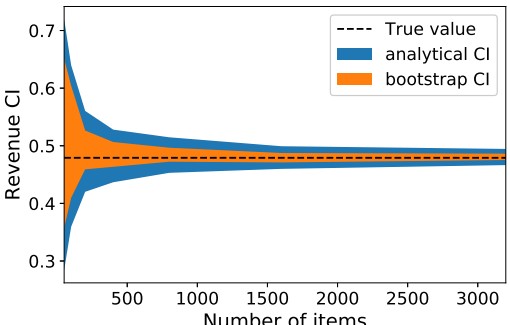

**Figure 6:** Revenue confidence intervals as a function of the number of items in fully synthetic experiments. The analytic CI comes from Eq. (22). The true value is the debiased surrogates for revenue in the limit market with interference $\mathcal{M}_\alpha$.

