# OpenReview forum: "Interference Among First-Price Pacing Equilibria: A Bias and Variance Analysis"
_ICLR.cc/2025/Conference — ICLR 2025 Poster_

### Official Review · Reviewer_pX7g · 2024-11-02

**Soundness:** 3
**Presentation:** 3
**Contribution:** 3
**Rating:** 8
**Confidence:** 3

**Summary:**

The paper proposes an estimation technique for A/B test revenue by modeling budget interference between two experiments as a mixture of distributions.

At first, the paper is hard to follow for readers unfamiliar with the FPPE formalism, such as myself. Even though I am familiar with advertising systems and budget pacing mechanisms. A lot of symbols and concepts to grasp. However, once I familiarized myself with the relevant cited background, the explanation of modeling of budget interference using a mixture of distribution makes sense, and I believe it's an important concept that this paper introduces.

The experimental section is quite shallow, but I believe that since the paper is focused on rigorous theory, extensive experiments are less important in such a paper.

However, there are many weaknesses of this paper, described below in the weaknesses section, that make me believe this paper requires additional work to make it ready for publication. The largest problem is, I believe, that the theoretical framework of an equilibrium does not appear to model real-world marketplaces with ever-changing pacing factors. However, the paper presents the technique as something applicable in practice, and no attempt is made to close this gap. More weaknesses appear in the weakness section.

**Strengths:**

- aims to solve an extremely important practical problem
- Introduces an interesting theoretical framework of modeling "budget interference" using a mixture of distributions.
- makes a good job explaining *why* such a modeling is reasonable.

**Weaknesses:**

- real budget pacing systems, from my experience, do not operate with constant pacing factors. But the FPPE theory assumes each buyer has one specific pacing factor defined by the equilibrium. There is no explanation in the paper for how this framework actually models the real world of dynamically changing pacing factors, and this limits the usefulness of the estimators in practice. If it is possible to apply the framework to such a dynamic system, there is no explanation in the paper for *how* to do it. So overall, there is no motivation explaining why the equilibrium framework is useful.
- the notation is extremely hard to follow. Typically there is a difference between how vectors, scalars, sets, random variables, and matrices are denoted. For example, vectors by boldface, matrices as upcase-bold, and so on.. It requires a lot of mental effort to follow all the symbols and concepts in the paper, which makes it practically unreadable to audience unfamiliar with the FPPE framework
- a paper aims to show us how to debias the revenue metrics in A/B tests, but it does *not* derive a debiased estimator for the revenue. only for the budget pacing factors. For such an important concept, I'd assume there should be at least a corollary for how the revenue estimator can be computed from the estimated pacing factors, and an explanation for why this estimator is also debiased (it's not obvious that a function of a de-biased estimator is also de-biased in the same sense).


## Update
Most of the weaknesses have been addressed by the rebuttal and the revised version.

**Questions:**

- is the decomposition into submarkets part of the contribution or not? It appears as something important, but it's in the introduction, rather than being in the main paper. Why? Please clarify this. If this is a contribution, it should be explicitly evident and not appear in the introduction. If it is not, this should be explicit in the paper, maybe with proper citations.
- Please explain - why is the equilibrium framework applicable in practice, given the changing nature of pacing factors?

---

> ### Author Response · Authors · 2024-11-22
> **Thank you!**
>
> > At first, the paper is hard to follow for readers unfamiliar with the FPPE formalism, such as myself. Even though I am familiar with advertising systems and budget pacing mechanisms. A lot of symbols and concepts to grasp. However, once I familiarized myself with the relevant cited background, the explanation of modeling of budget interference using a mixture of distribution makes sense, and I believe it's an important concept that this paper introduces.
>
>
> Thank you for carefully reading the model, we’ll be adding an additional notation table to aid in the readability.
>
>
> > Real budget pacing systems, from my experience, do not operate with constant pacing factors. But the FPPE theory assumes each buyer has one specific pacing factor defined by the equilibrium. There is no explanation in the paper for how this framework actually models the real world of dynamically changing pacing factors, and this limits the usefulness of the estimators in practice. If it is possible to apply the framework to such a dynamic system, there is no explanation in the paper for how to do it. So overall, there is no motivation explaining why the equilibrium framework is useful.
>
>
>
>
> We agree with the reviewer that real-world pacing systems are typically implemented using a pacing multiplier as a control that’s increased or decreased based on whether the campaign is currently overspending or underspending [1,2]. The FPPE model captures the steady state that these dynamics attempt to converge to. While we agree that capturing the dynamics of changing pacing multipliers is of interest, we also think that this is a very hard problem. Concretely, we are not aware of any paper that rigorously captures statistical inference under a dynamic pacing model, even without the type of interference that we study here. To that end, we think that our paper makes an important step towards modeling interference in segmented A/B testing, while leaving open how to handle the dynamics of pacing multiplier updates.
>
>
> [1] Balseiro, Gur. “Learning in repeated auctions with budgets: Regret minimization and equilibrium” Management Science 2019.
> [2] Conitzer et al. “Multiplicative pacing equilibria in auction markets” Operations Research 2022.
>
> > The notation is extremely hard to follow. Typically there is a difference between how vectors, scalars, sets, random variables, and matrices are denoted. For example, vectors by boldface, matrices as uppercase-bold, and so on.. It requires a lot of mental effort to follow all the symbols and concepts in the paper, which makes it practically unreadable to audience unfamiliar with the FPPE framework
>
> Thank you for the feedback. We will revise the notation and create a table of notations in order to alleviate this issue.
>
> > The paper aims to show us how to debias the revenue metrics in A/B tests, but it does not derive a debiased estimator for the revenue. only for the budget pacing factors. For such an important concept, I'd assume there should be at least a corollary for how the revenue estimator can be computed from the estimated pacing factors, and an explanation for why this estimator is also debiased (it's not obvious that a function of a de-biased estimator is also de-biased in the same sense).
>
> In Appendix E, we actually do develop a debiasing theory for revenue, similar to the one presented for pacing multipliers in the body: we construct a debiased surrogate, and show asymptotic normality and confidence intervals for that surrogate. In a technical sense, revenue is a smooth function of the pacing multipliers, and so the relevant debiasing theory can be developed somewhat straightforwardly from our pacing multiplier results. For this reason and space constraint reasons, we put the revenue theory in the appendix. We will update the paper to make it clearer that we have this theory.
>
> > Is the decomposition into submarkets part of the contribution or not? It appears as something important, but it's in the introduction, rather than being in the main paper. Why? Please clarify this. If this is a contribution, it should be explicitly evident and not appear in the introduction. If it is not, this should be explicit in the paper, maybe with proper citations.
>
> As far as we know, the submarket decomposition idea has not been published anywhere, so we do not have anyone to cite for it. At the same time, we would not be surprised if it has been implemented by others in addition to us, since it’s a simple idea. Thus, we do not necessarily want to claim it as a contribution. We can state something to this effect in the paper to make it clearer regarding the provenance of the idea.
>
> > Please explain - why is the equilibrium framework applicable in practice, given the changing nature of pacing factors?
>
> Addressed in the section on Weaknesses above.

---

> ### Comment · Reviewer_pX7g · 2024-11-24
> **Some comments**
>
> ## Regarding notation
> Personally, I believe that a consistent notation, along the lines that is demonstrated in the ICLR template (in the math\_commands.tex file), to differentiate between random variables, vectors, matrices, sets, etc... contribute much more than a notation table. The FPPE formalism has a lot of concepts tied together in one definition, and it's much easier to grasp them when the *types* of the objects involved are clear.
>
> ## Regarding factor dynamics
> It may have not been clear from the original review, but I think that the main issue with factor dynamics is ignoring them. The claim of the paper is to give a practical tool, yet the paper give a tool which appears only theoretical in nature, since factor dynamics are not addressed in any way. The theoretical contribution is clear, it's important, and appears to be sound. But the paper's claim is to do more than that.
>
> So if the authors chose to remain with the claim that their contribution is also of practical nature, either there has to be some arguments for why this is also applicable in practice (i.e. more extensive experiments or simulations in practice), or there has to be a theoretical argument explaining the applicability in practice, such as "if the dynamics are 'close' to the equilibrium up to $\varepsilon$, then our estimator's error is increased by at most $f(\varepsilon)$". At its current state, the contribution appears not to conform to what is claimed.

---

> > ### Comment · Reviewer_pX7g · 2024-11-26
> > **Score change**
> >
> > I read the revised version. The notation change and the clarifications make the paper easier to follow. My concern about revenue debiasing has been resolved by the comments and the clarification. And the paper doesn't seem to claim more than it does. I have increased my scores accordingly, and I recommend accepting the paper (a score of 8).

---

### Official Review · Reviewer_Ms53 · 2024-11-03

**Soundness:** 3
**Presentation:** 1
**Contribution:** 2
**Rating:** 5
**Confidence:** 3

**Summary:**

This paper studies how to estimate market equilibria in each submarket when there is interference across submarkets. The authors consider the first-price pacing equilibrium (FPPE) model where each buyer uses a single pacing multiplier to shade values. For each submarket, some items outside would also attract the buyers in this submarket, which is unavoidable since we are unable to find completely isolated submarkets. From the modeling point of view, the supply is contaminated by another distribution. The authors first propose a debiased surrogate, which can approximate the limit FPPE in the uncontaminated market based on the limit FPPE in the contaminated market, and then prove that it has only a small error due to the removal of the first-order bias. Then, since the finit FPPE will converge to the limit FPPE in probability, the above estimator can be applied to the actual dynamic market with a slight modification. The authors also present two asymptotic normality results to further demonstrate the superiority of the estimator. Experiments on semi-synthetic data show that the proposed estimator is indeed less biased in terms of pacing multipliers and revenue.

**Strengths:**

1. I can understand that A/B test design is indeed a very important issue in the industry. Since buyers will use strategic behaviors to counter the platform's strategy or mechanism, a/b testing is the most effective way to test the platform's strategy/mechanism. However, the vanilla a/b test setup will bring about the issue of mutual influence between a/b groups, which may interfere with the experimental results.

2. The theoretical proof in the paper is non-trivial and very solid. The authors have proved from many aspects that the estimator can indeed converge well to the limit FPPE in the uncontaminated market, which the platform hopes to observe.

**Weaknesses:**

Overall, this paper is hard to parse. The flow of this paper is not clear enough to assist readers well. The abstract and introduction of this paper both start with a/b testing, but the contribution part does not mention it. It was not until page 6 that I began to realize what the authors are doing. After reading the paper several times, I finally understood the authors' real contribution.

According to my understanding, the real question is how to study the impact of different strategies/mechanisms on market FPPE. The method proposed by the authors is to find relatively isolated submarkets as a/b groups, and then construct an estimator to eliminate cross-market interference. As far as I know, the former is a method that has been adopted in the industry (of course, the issue of cross-market interference remains), while the latter is tailor-made for the FPPE model. Therefore, the focus of this paper's contribution should actually be on FPPE rather than ab test design. In fact, even if all ab test parts are removed, this will still be a complete paper that studies how to approximate the limit FPPE of an uncontaminated market using contaminated data.

Since the estimator in this paper cannot be directly extended to other scenarios, I suggest that the authors should not emphasize the proposal of a novel ab test framework, but use the research on FPPE as the background to introduce challenges and solutions.

**Questions:**

NA

---

> ### Author Response · Authors · 2024-11-22
> **Thank you for your review!**
>
> > Overall, this paper is hard to parse. The flow of this paper is not clear enough to assist readers well. The abstract and introduction of this paper both start with a/b testing, but the contribution part does not mention it. It was not until page 6 that I began to realize what the authors are doing. After reading the paper several times, I finally understood the authors' real contribution.
>
> First, thank you for reading the paper carefully and repeatedly! We agree that the paper is somewhat hard to parse. We will try our best to rectify this, using some combination of the following ideas:
>
> - We will add a paragraph in the beginning that more explicitly lays out the relationship between the AB testing, the contaminated supply framework, and how we eventually model submarket AB testing.
> - We will add a notation table
>
> We’re additionally open to suggestions from the reviewer on any other improvements you would suggest.
>
>
> > According to my understanding, the real question is how to study the impact of different strategies/mechanisms on market FPPE. The method proposed by the authors is to find relatively isolated submarkets as a/b groups, and then construct an estimator to eliminate cross-market interference. As far as I know, the former is a method that has been adopted in the industry (of course, the issue of cross-market interference remains), while the latter is tailor-made for the FPPE model. Therefore, the focus of this paper's contribution should actually be on FPPE rather than ab test design. In fact, even if all ab test parts are removed, this will still be a complete paper that studies how to approximate the limit FPPE of an uncontaminated market using contaminated data. Since the estimator in this paper cannot be directly extended to other scenarios, I suggest that the authors should not emphasize the proposal of a novel ab test framework, but use the research on FPPE as the background to introduce challenges and solutions.
>
> We agree that our theoretical model could be presented on its own as a general theory, and indeed the results could be used to model other “contamination” scenarios than the AB testing scenario. However, we view the AB test design setting as the most important motivator for the design of our theoretical results, and thus we disagree regarding your suggested de-emphasizing of AB testing. We do agree that *general* AB testing is not addressed by our framework, but we are not trying to address general AB testing, we are specifically trying to address AB testing under budget-management induced interference. For that, we believe the FPPE model is the only known model for capturing these effects in a way where we can tractably hope to give statistical guarantees.
>
> Of course we agree that our results are not generalizable to AB testing outside of budget management settings. But budget management systems are prevalent in practice, and they induce a particular form of interference. For this reason, we think that it is important to study how one can attempt to address interference introduced by these systems. As far as we know, FPPE offers the only theoretical model under which we can perform this sort of analysis given the current literature, due to the mathematical and computational difficulty involved under other models of budget management.
>
> In writing the above response, we recognized that this may not be the notion of generalizability that you intended, so if you had something else in mind then please let us know.

---

### Official Review · Reviewer_5WoS · 2024-11-10

**Soundness:** 3
**Presentation:** 3
**Contribution:** 3
**Rating:** 8
**Confidence:** 3

**Summary:**

This paper considers A/B testing in online marketplaces, where interference arises because items can be recommended to advertisers in both the control and the treatment groups. The paper adopts the first-price pacing equilibrium (FPPE) framework from prior work [Conitzer et al., 2022], and analyzes how the equilibrium (i.e., pacing parameters \beta and the total revenue) changes as a function of the level of contamination/interference. The paper proposes a first-order bias correction by Taylor expansion and proves consistency and asymptotic normality guarantees under certain regularity conditions. The paper conducts semi-synthetic experiments to demonstrate the effectiveness of the proposed estimators (where the distributions come from real data).

**Strengths:**

Interference is a well-known, important, and practical problem in A/B testing of online marketplaces. The paper meaningfully extends prior work on pacing equilibria by considering how interference affects the results. The paper provides rigorous theoretical guarantees, complemented by preliminary semi-synthetic experiments.

**Weaknesses:**

While I find the formulation and assumptions proposed in the paper reasonable, I think the paper can benefit from a discussion on limitations. For example, a natural alternative model is not to partition items into a good set and a bad set, but instead assume an item can be "good" for some groups but "bad" (interference) for other groups. The paper may also discuss other types of interference that the proposed approach does not cover, etc.

===
Additional minor comments:

1. I think the claim in the abstract on demonstrating "the effectiveness of this approach on real experiments on advertising markets at Meta" is an overstatement. I think the paper should make it clear that only semi-synthetic experiments are performed.

2. While I understand citing unpublished work is optional, I think the paper can benefit from citing the following paper:
Zhu, Cai, Zheng, and Si. "Seller-Side Experiments under Interference Induced by Feedback Loops in Two-Sided Platforms". arXiv, 2024.

3.  I find the term "buyer" confusing, as "buyers" can be understood as users who make purchases or sellers who bid for ads. I prefer the terminology of advertisers vs. users introduced in other parts of the paper.

4.  I don't understand details of the experiment in Fig 1. What're the axes? Are these ratios of the two experimental designs? How are two experiments paired? What is the guardrail metric? Also a 79% agreement compared to the 81.5% optimal agreement appears pretty good to me. Is the paper trying to address the remaining 2.5% of the cases?

5.  In Paragraph L87-118, two graphs are introduced. One is a bipartite graph between advertisers and users, from which a graph for advertisers is derived for clustering ("the edge weight between a pair of advertisers..."). It would be helpful to clarify they are not the same graph.

6.  Paragraph L54-63: "Sec" -> "Fig"

**Questions:**

1. The theoretical guarantees heavily rely on the parameter \eta for the error in estimating the Hessian. Could the authors comment on the the rate \eta for the proposed finite differencing Hessian estimator proposed in the paper?

2.  Does the guarantee on the revenue in Theorem 4 generalize to estimating other quantities that are a functional of the parameters \beta?

3. Providing a discussion on limitations and addressing my minor comment #4 above would be helpful.

---

> ### Author Response · Authors · 2024-11-22
> **Thank you for your review!**
>
> > I think the claim in the abstract on demonstrating "the effectiveness of this approach on real experiments on advertising markets at Meta" is an overstatement. I think the paper should make it clear that only semi-synthetic experiments are performed.
>
> We will make it clear in the paper. We changed the statement to "the effectiveness of this approach on semi-synthetic experiments created based on advertising markets at Meta".
>
> > While I understand citing unpublished work is optional, I think the paper can benefit from citing the following paper: Zhu, Cai, Zheng, and Si. "Seller-Side Experiments under Interference Induced by Feedback Loops in Two-Sided Platforms". arXiv, 2024.
>
> We thank you for pointing out this paper which also addresses interference issues in pacing platforms. We added the following citation.
>
> “The recent work by Zhu et al investigates the effects of interference caused by feedback loops, which are prevalent in seller-side experiments, recommendation systems, and pacing systems. They specifically focus on counterfactual interleaving design, formulate the interference, and theoretically estimate its impact.”
>
> > I find the term "buyer" confusing, as "buyers" can be understood as users who make purchases or sellers who bid for ads. I prefer the terminology of advertisers vs. users introduced in other parts of the paper.
>
>
> We will explain this clearly. Thanks for pointing out this ambiguity.
>
>
> > I don't understand the details of the experiment in Fig 1. What're the axes? Are these ratios of the two experimental designs? How are two experiments paired? What is the guardrail metric? Also a 79% agreement compared to the 81.5% optimal agreement appears pretty good to me. Is the paper trying to address the remaining 2.5% of the cases?
>
>
> The x-axis represents the treatment effect in a full-market budget-constrained A/B test, expressed as a ratio of test group value to control group value. The y-axis represents the treatment effect in a sub-market budget-constrained A/B test, also expressed as a ratio of test group value to control group value. Two experiments are paired if they run in the same time period, on non-overlapping user groups, and with identical treatment.
>
> The guardrail metric is “impression shift at infra-day frequency”: when interference occurs, one treatment group will “steal” impressions from the other group and due to the nature of pacing algorithms, these impressions shifts between treatment groups usually exhibit some diurnal pattern, say impressions flow from test to control in the daytime and the other way around at night. By measuring the distance between the two impression curves, we can detect interference bias.
>
> We agree that the sign consistency is not a concern even without the methodology of the paper. However, the main contribution is to remove the remaining bias in the magnitude of the treatment effect estimate. As you can see in the plot, while the sign agreement is high, the points are typically off the diagonal. To address this magnitude error, the theory in later sections is necessary.
>
> > In Paragraph L87-118, two graphs are introduced. One is a bipartite graph between advertisers and users, from which a graph for advertisers is derived for clustering ("the edge weight between a pair of advertisers..."). It would be helpful to clarify they are not the same graph.
>
>
> We added a footnote explaining the distinction.
>
>
> > The theoretical guarantees heavily rely on the parameter \eta for the error in estimating the Hessian. Could the authors comment on the rate \eta for the proposed finite differencing Hessian estimator proposed in the paper?
>
> We have two theorems for different error rates $ \eta_t $ (Theorem 3). If $ \eta_t $ is estimated at the rate $ o(1/\sqrt t) $, which can be achieved by using a separate set of larger historical data, then the normality of our unbiased pacing multiplier estimator holds without further assumptions. If the bidgap condition holds additionally, then the Hessian expression has a simplified form, and can be estimated easily at the rate $ O(1/\sqrt t) $, and the normality of our unbiased pacing multiplier holds.
>
> We propose in practice we only estimate the diagonal part of the Hessian. We used this in experiments and it is scalable and performant.
>
> > Does the guarantee on the revenue in Theorem 4 generalize to estimating other quantities that are a functional of the parameters \beta?
>
>
> Yes. Our debiased procedure can be applied to estimate any smooth function $\phi$ of the limit pacing multiplier.
>
> > Providing a discussion on limitations and addressing my minor comment #4 above would be helpful.
>
>
> Addressed above.

---

> > ### Comment · Reviewer_5WoS · 2024-11-27
> >
> > Thank you for the response. I have read it and it has addressed most of my questions (my questions are not major critiques anyways).
> >
> > A few additional minor comments:
> > - Regarding Fig. 1, it would be helpful to motivate (briefly in writing) the importance of accurately estimating the magnitude of the treatment effect (if the sign is correct most of the times without correction). In practice, for business decisions, the sign is often more critical than the precise magnitude.
> > - As I mentioned in my original review, it would be helpful to provide a thorough discussion on the limitation of the work, for example, on what specific scenarios and interferences the proposed model does not capture.

---

### Meta-Review · Area_Chair_sC9n · 2024-12-22

**Metareview:**

This paper studies A/B testing with interference in online marketplaces. The authors considers first-price pacing equilibrium (FPPE) and analyzes how the equilibrium changes according to contamination/interference. The authors then proposes a debias surrogate against the first-order bias and an estimator. The reviewers recognized the following strengths:
- The problem of interference in A/B testing is important and practical.
- Theoretical analysis is a major contribution. The results and proofs are non-trivial and sound.
- Semi-synthetic experiments demonstrated the effectiveness of the proposed estimators.

Weaknesses:
-  Presentation issue: there are concerns about unclear notations, confusing concepts, and missing details of experiments. The revision with clarifications addressed the issue.
- Limited generalization ability: the result is not generalizable to the problem outside of budget management settings.

After discussion with authors and among reviewers, all reviewers agree to accept the paper considering the importance of the problem and the theoreical contribution. I agree with the reviewers and recommend acceptance.

**Additional Comments On Reviewer Discussion:**

The following weaknesses were identified:
-  Presentation issue: there are concerns about unclear notations, confusing concepts, and missing details of experiments. The revision with clarifications addressed the issue (Reviewer 5WoS and pX7g). The revision and clarification leads to increased score by Reviewer pX7g.
- Limited generalization ability: the result is not generalizable to the problem outside of budget management settings. This problem raised by Reviewer Ms53 is not addressed and the reviewer did not change the rating. However, the reviewer agrees to accept the paper during discussion consider the importance of the problem and the theoretical analysis.

---

### Decision · Program_Chairs · 2025-01-22

Accept (Poster)